# Discrete-Valued Neural Communication in Structured Architectures Enhances Generalization

**Dianbo Liu**[*]
Mila

**Alex Lamb**[*]
Mila

**Kenji Kawaguchi**
Harvard University

**Anirudh Goyal**
Mila

**Chen Sun**
Mila

**Michael C. Mozer**
Google Research, Brain Team

**Yoshua Bengio**
Mila

## Abstract

Deep learning has advanced from fully connected architectures to structured models organized into components, e.g., the transformer composed of positional elements, modular architectures divided into slots, and graph neural nets made up of nodes. The nature of structured models is that communication among the components has a bottleneck, typically achieved by restricted connectivity and attention. In this work, we further tighten the bottleneck via discreteness of the representations transmitted between components. We hypothesize that this constraint serves as a useful form of inductive bias. Our hypothesis is motivated by past empirical work showing the benefits of discretization in non-structured architectures as well as our own theoretical results showing that discretization increases noise robustness and reduces the underlying dimensionality of the model. Building on an existing technique for discretization from the VQ-VAE, we consider multi-headed discretization with shared codebooks as the output of each architectural component. One motivating intuition is human language in which communication occurs through multiple discrete symbols. This form of communication is hypothesized to facilitate transmission of information between functional components of the brain by providing a common interlingua, just as it does for human-to-human communication. Our experiments show that *discrete-valued neural communication (DVNC)* substantially improves systematic generalization in a variety of architectures—transformers, modular architectures, and graph neural networks. We also show that the DVNC is robust to the choice of hyperparameters, making the method useful in practice.

## 1 Introduction

In AI, there has long been a tension between subsymbolic and symbolic architectures. Subsymbolic architectures, like neural networks, utilize continuous representations and statistical computation. Symbolic architectures, like production systems (Laird et al., 1986) and traditional expert systems, use discrete, structured representations and logical computation. Each architecture has its strengths: subsymbolic computation is useful for perception and control, symbolic computation for higher level, abstract reasoning. A challenge in integrating these approaches is developing unified learning procedures.

As a step toward bridging the gap, recent work in deep learning has focused on constructing structured architectures with multiple components that interact with one another. For instance, graph neural networks are composed of distinct nodes (Kipf et al., 2019; Scarselli et al., 2008; Kipf et al., 2018;

---

[*]co-first author. Address correspondence to `liudianbo@gmail.com`, `alex6200@gmail.com`, and `kkawaguchi@fas.harvard.edu`.

35th Conference on Neural Information Processing Systems (NeurIPS 2021).

Santoro et al., 2017; Raposo et al., 2017; Bronstein et al., 2017; Gilmer et al., 2017; Tacchetti et al., 2018; Van Steenkiste et al., 2018), transformers are composed of positional elements (Bahdanau et al., 2014; Vaswani et al., 2017), and modular models are divided into slots or modules with bandwidth limited communication (Jacobs et al., 1991; Bottou and Gallinari, 1991; Goyal and Bengio, 2020; Ronco et al., 1997; Reed and De Freitas, 2015; Lamb et al., 2021; Andreas et al., 2016; Rosenbaum et al., 2017; Fernando et al., 2017; Shazeer et al., 2017; Rosenbaum et al., 2019; Kucinski et al., 2021).

Although these structured models exploit the discreteness in their architectural components, the present work extends these models to leverage discreteness of representations, which is an essential property of symbols. We propose to learn a common *codebook* that is shared by all components for inter-component communication. The codebook permits only a discrete set of communicable values. We hypothesize that communication based on the use and re-use of discrete symbols will provide two benefits:

- Discrete symbols limit the bandwidth of representations whose interpretation needs to be learned and synchronized across modules. It may therefore serve as a common language for interaction and facilitate learning.
- The use of shared discrete symbols will promote systematic generalization by allowing for the re-use of previously encountered symbols in new situations. This makes it easier to hot-swap one component for another when new out-of-distribution (OOD) settings arise that require combining existing components in novel ways.

Our work is inspired by cognitive science, neuroscience, and mathematical considerations. From the cognitive science perspective, we can consider different components of structured neural architectures to be analogous to autonomous agents in a distributed system whose ability to communicate stems from sharing the same language. If each agent speaks a different language, learning to communicate would be slow and past experience would be of little use when the need arises to communicate with a new agent. If all agents learn the same language, each benefits from this arrangement. To encourage a common language, we limit the expressivity of the vocabulary to discrete symbols that can be combined combinatorially (Tomasello, 2009). From the neuroscience perspective, we note that various areas in the brain, including the hippocampus (Sun et al., 2020; Quiroga et al., 2005; Wills et al., 2005), the prefrontal cortex (Fujii and Graybiel, 2003), and sensory cortical areas (Tsao et al., 2006) are tuned to discrete variables (concepts, actions, and objects), suggesting the evolutionary advantage of such encoding, and its contribution to the capacity for generalization in the brain. From a theoretical perspective, we present analysis suggesting that multi-head discretization of inter-component communication increases model sensitivity and reduces underlying dimensions (Section 2). These sources of inspiration lead us to the proposed method of *Discrete-Valued Neural Communication (DVNC)*.

Architectures like graph neural networks (GNNs), transformers, and slot-based or modular neural networks consist of articulated specialist components, for instance, nodes in GNNs, positions in transformers, and slots/modules for modular models. We evaluate the efficacy of DVNC in GNNs, transformers, and in a modular recurrent architecture called RIMs. For each of these structured architectures, we keep the original architecture and all of its specialist components the same. The only change is that we impose discretization in the communication between components (Figure 1).

Our work is organized as follows. First, we introduce DVNC and present theoretical analysis showing that DVNC improves sensitivity and reduces metric entropy of models (the logarithm of the covering number). Then we explain how DVNC can be incorporated into different model architectures. And finally we report experimental results showing improved OOD generalization with DVNC.

## 2 Discrete-Value Neural Communication and Theoretical Analysis

In this section, we begin with the introduction of Discrete-Value Neural Communication (DVNC) and proceed by conducting a theoretical analysis of DVNC affects the sensitivity and metric entropy of models. We then explain how DVNC can be used within several different architectures.

**Discrete-Value Neural Communication (DVNC)**    The process of converting data with continuous attributes into data with discrete attributes is called discretization (Chmielewski and Grzymala-Busse,

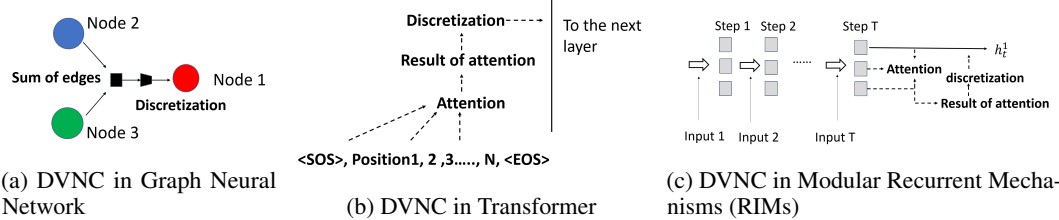

(a) DVNC in Graph Neural Network

(b) DVNC in Transformer

(c) DVNC in Modular Recurrent Mechanisms (RIMs)

Figure 1: Communication among different components in neural net models is discretized via a shared codebook. In modular recurrent neural networks and transformers, values of results of attention are discretized. In graph neural networks, communication from edges is discretized.

1996). In this study, we use discrete latent variables to quantize information communicated among different modules in a similar manner as in Vector Quantized Variational AutoEncoder (VQ-VAE) (Oord et al., 2017). Similar to VQ-VAE, we introduce a discrete latent space vector $e \in \mathbb{R}^{L \times (m/G)}$ where $L$ is the size of the discrete latent space (i.e., an $L$-way categorical variable), and $m$ is the dimension of each latent embedding vector $e_j$. Here, $L$ and $m$ are both hyperparameters. In addition, by dividing each target vector into $G$ segments or discretization heads, we separately quantize each head and concatenate the results (Figure 2). More concretely, the discretization process for each vector $h \in \mathcal{H} \subset \mathbb{R}^m$ is described as follows. First, we divide a vector $h$ into $G$ segments $s_1, s_2, \ldots, s_G$ with $h = \text{CONCATENATE}(s_1, s_2, \ldots, s_G)$, where each segment $s_i \in \mathbb{R}^{m/G}$ with $\frac{m}{G} \in \mathbb{N}^+$. Second, we discretize each segment $s_i$ separately:

$$e_{o_i} = \text{DISCRETIZE}(s_i), \quad \text{where } o_i = \underset{j \in \{1, \ldots, L\}}{\arg\min} \|s_i - e_j\|.$$

Finally, we concatenate the discretized results to obtain the final discretized vector $Z$ as

$$Z = \text{CONCATENATE}(\text{DISCRETIZE}(s_1), \text{DISCRETIZE}(s_2), \ldots, \text{DISCRETIZE}(s_G)).$$

The multiple steps described above can be summarized by $Z = q(h, L, G)$, where $q(\cdot)$ is the whole discretization process with the codebook, $L$ is the codebook size, and $G$ is number of segments per vector. It is worth emphasizing that the codebook $e$ is shared across all communication vectors and heads, and is trained together with other parts of the model.

The overall loss for model training is:

$$\mathcal{L} = \mathcal{L}_{\text{task}} + \frac{1}{G} \left( \sum_i^G \| \text{sg}(s_i) - e_{o_i} \|_2^2 + \beta \sum_i^G \| s_i - \text{sg}(e_{o_i}) \|_2^2 \right) \tag{1}$$

where $\mathcal{L}_{\text{task}}$ is the loss for specific task, *e.g.*, cross entropy loss for classification or mean square error loss for regression, sg refers to a stop-gradient operation that blocks gradients from flowing into its argument, and $\beta$ is a hyperparameter which controls the reluctance to change the code. The second term $\sum_i^G \| \text{sg}(s_i) - e_{o_i} \|_2^2$ is the codebook loss, which only applies to the discrete latent vector and brings the selected $e_{o_i}$ close to the output segment $s_i$. The third term $\sum_i^G \| s_i - \text{sg}(e_{o_i}) \|_2^2$ is the commitment loss, which only applies to the target segment $s_i$ and trains the module that outputs $s_i$ to make $s_i$ stay close to the chosen discrete latent vector $e_{o_i}$. We picked $\beta = 0.25$ as in the original VQ-VAE paper (Oord et al., 2017). We initialized $e$ using $k$-means clustering on vectors $h$ with $k = L$ and trained the codebook together with other parts of the model by gradient descent. When there were multiple $h$ vectors to discretize in a model, the mean of the codebook and commitment loss across all $h$ vectors was used. Unpacking this equation, it can be seen that we adapted the vanilla VQ-VAE loss to directly suit our discrete communication method (Oord et al., 2017). In particular, the VQ-VAE loss was adapted to handle multi-headed discretization by summing over all the separate discretization heads.

In the next subsection, we use the following additional notation. The function $\phi$ is arbitrary and thus can refer to the composition of an evaluation criterion and the rest of the network following discretization. Given any function $\phi : \mathbb{R}^m \to \mathbb{R}$ and any family of sets $S = \{S_1, \ldots, S_K\}$ with $S_1, \ldots, S_K \subseteq \mathcal{H}$, we define the corresponding function $\phi_k^S$ by $\phi_k^S(h) = \mathbb{1}\{h \in S_k\}\phi(h)$ for all $k \in [K]$, where $[K] = \{1, 2, \ldots, K\}$. Let $e \in E \subset \mathbb{R}^{L \times m}$ be fixed and we denote by $(Q_k)_{k \in [L^G]}$ all the possible values after the discretization process: i.e., $q(h, L, G) \in \cup_{k \in [L^G]}\{Q_k\}$ for all $h \in \mathcal{H}$.

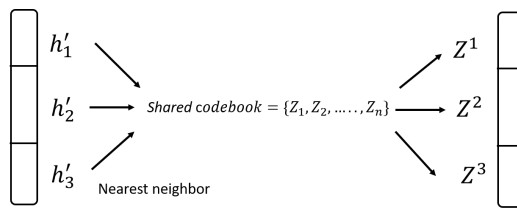

Figure 2: In structured architectures, communication is typically vectorized. In DVNC, this communication vector is first divided into discretization heads. Each head is discretized separately to the nearest neighbor of a collection of latent codebook vectors which is shared across all the heads. The discretization heads are then concatenated back into the same shape as the original vector.

Table 1: Communication with discretized values achieves a noise-sensitivity bound that is independent of the number of dimensions $m$ and network lipschitz constant $\bar{\varsigma}_k$ and only depends on the number of discretization heads $G$ and codebook size $L$.

| Communication Type | Example | Sensitivity Bounds (Thm 1, 2) |
|---|---|---|
| Communication with continuous signals is expressive but can take a huge range of novel values, leading to poor systematic generalization | $m \sim 10^5$ | $\mathcal{O}\left(\sqrt{\frac{m\ln(4\sqrt{nm})+\ln(2/\delta)}{2n}} + \frac{\bar{\varsigma}_k R_\mathcal{H}}{\sqrt{n}}\right)$ |
| Communication with multi-head discrete-values is both expressive and sample efficient | "John owns a car" $G = 15$ $L = 30$ | $\mathcal{O}\left(\sqrt{\frac{G\ln(L)+\ln(2/\delta)}{2n}}\right)$ |

**Theoretical Analysis**   This subsection shows that adding the discretization process has two potential advantages: (1) it improves noise sensitivity or robustness and (2) it reduces metric entropy. These are proved in Theorems 1–2, illustrated by examples (Table 1), and explored in analytical experiments using Gaussian-distributed vectors (Figure 3).

To understand the advantage on noise sensitivity or robustness, we note that there is an additional error incurred by noise *without discretization*, i.e., the second term $\bar{\varsigma}_k R_\mathcal{H}/\sqrt{n} \geq 0$ in the bound of Theorem 2 ($\bar{\varsigma}_k$ and $R_\mathcal{H}$ are defined in Theorem 2). This term is defined to be the noise sensitivity (or robustness) term. This term due to noise disappears *with discretization* in the bound of Theorem 1 as the discretization process reduces the sensitivity to noise. This is because the discretization process lets the communication become invariant to noise within the same category; e.g., the communication is invariant to different notions of "cats".

To understand the advantage of discretization on dimensionality, we can see that it reduces the metric entropy of $m\ln(4\sqrt{nm})$ *without discretization* (in Theorem 2) to that of $G\ln(L)$ *with discretization* (in Theorem 1). As a result, the size of the codebook $L$ affects the underlying dimension in a weak (logarithmic) fashion, while the number of dimensions $m$ and the number of discretization heads $G$ scale the underlying dimension in a linear way. Thus, the discretization process successfully lowers the metric entropy for any $n \geq 1$ as long as $G\ln(L) < m\ln(4\sqrt{m})$. This is nearly always the case as the number of discretization heads $G$ is almost always much smaller than the number of units $m$. Intuitively, a discrete language has combinatorial expressiveness, making it able to model complex phenomena, but still lives in a much smaller space than the world of unbounded continuous-valued signals (as $G$ can be much smaller than $m$).

**Theorem 1.** (with discretization) *Let $S_k = \{Q_k\}$ for all $k \in [L^G]$. Then, for any $\delta > 0$, with probability at least $1 - \delta$ over an iid draw of $n$ examples $(h_i)_{i=1}^n$, the following holds for any $\phi : \mathbb{R}^m \to \mathbb{R}$ and all $k \in [L^G]$: if $|\phi_k^S(h)| \leq \alpha$ for all $h \in \mathcal{H}$, then*

$$\left| \mathbb{E}_h[\phi_k^S(q(h, L, G))] - \frac{1}{n}\sum_{i=1}^n \phi_k^S(q(h_i, L, G)) \right| = \mathcal{O}\left(\alpha\sqrt{\frac{G\ln(L)+\ln(2/\delta)}{2n}}\right), \qquad (2)$$

*where no constant is hidden in $\mathcal{O}$.*

**Theorem 2.** (without discretization) *Assume that $\|h\|_2 \leq R_\mathcal{H}$ for all $h \in \mathcal{H} \subset \mathbb{R}^m$. Fix $\mathcal{C} \in \operatorname{argmin}_{\bar{\mathcal{C}}}\{|\bar{\mathcal{C}}| : \bar{\mathcal{C}} \subseteq \mathbb{R}^m, \mathcal{H} \subseteq \cup_{c\in\bar{\mathcal{C}}}\mathcal{B}[c]\}$ where $\mathcal{B}[c] = \{x \in \mathbb{R}^m : \|x - c\|_2 \leq R_\mathcal{H}/(2\sqrt{n})\}$. Let $S_k = \mathcal{B}[c_k]$ for all $k \in [|\mathcal{C}|]$ where $c_k \in \mathcal{C}$ and $\cup_k\{c_k\} = \mathcal{C}$. Then, for any $\delta > 0$, with probability at least $1 - \delta$ over an iid draw of $n$ examples $(h_i)_{i=1}^n$, the following holds for any $\phi : \mathbb{R}^m \to \mathbb{R}$ and all $k \in [|\mathcal{C}|]$: if $|\phi_k^S(h)| \leq \alpha$ for all $h \in \mathcal{H}$ and $|\phi_k^S(h) - \phi_k^S(h')| \leq \varsigma_k\|h - h'\|_2$ for all $h, h' \in S_k$, then*

$$\left| \mathbb{E}_h[\phi_k^S(h)] - \frac{1}{n}\sum_{i=1}^n \phi_k^S(h_i) \right| = \mathcal{O}\left(\alpha\sqrt{\frac{m\ln(4\sqrt{nm})+\ln(2/\delta)}{2n}} + \frac{\bar{\varsigma}_k R_\mathcal{H}}{\sqrt{n}}\right), \qquad (3)$$

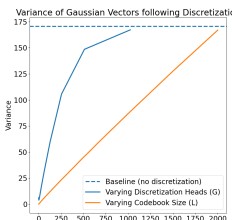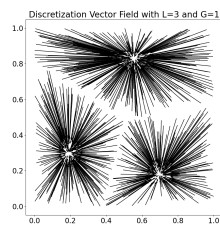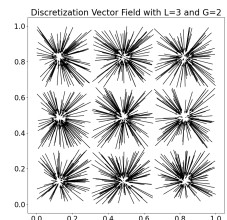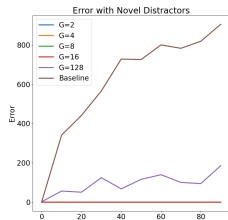

Figure 3: We perform empirical analysis on Gaussian vectors to build intuition for our theoretical analysis. Expressiveness scales much faster as we increase discretization heads than as we increase the size of the codebook. This can be seen when we measure the variance of a collection of Gaussian vectors following discretization (left), and can also be seen when we plot a vector field of the effect of discretization (center). Discretizing the values from an attention layer trained to select a fixed Gaussian vector makes it more robust to novel Gaussian distractors (right). For more details, see Appendix B.

where no constant is hidden in $\mathcal{O}$ and $\bar{\varsigma}_k = \varsigma_k \left( \frac{1}{n} \sum_{i=1}^n \mathbb{1}\{h_i \in \mathcal{B}[c_k]\} \right)$.

The two proofs of these theorems use the same steps and are equally tight as shown in Appendix D. Equation (3) is also tight as that of related work as discussed in Appendix C.2. The set $S$ is chosen to cover the original continuous space $\mathcal{H}$ in Theorem 2 (via the $\epsilon$-covering $\mathcal{C}$ of $\mathcal{H}$), and its discretized space in Theorem 1. Equations (2)–(3) hold for all functions $\phi : \mathbb{R}^m \to \mathbb{R}$, including the maps that depend on the samples $(h_i)_{i=1}^n$ via any learning processes. For example, we can set $\phi$ to be an evaluation criterion of the latent space $h$ or the composition of an evaluation criterion and any neural network layers that are learned with the samples $(h_i)_{i=1}^n$. In Appendix A, we present additional theorems, Theorems 3–4, where we analyze the effects of learning the map $x \mapsto h$ and the codebook $e$ via input-target pair samples $((x_i, y_i))_{i=1}^n$.

Intuitively, the proof shows that we achieve the improvement in sample efficiency when $G$ and $L$ are small, with the dependency on $G$ being significantly stronger (details in Appendix). Moreover, the dependency of the bound on the Lipschitz constant $\varsigma_k$ is eliminated by using discretization. Our theorems 1–2 are applicable to all of our models for recurrent neural networks, transformers, and graph neural networks (since the function $\phi$ is arbitrary) in the following subsections.

**Communication along edges of Graph Neural Network** One model where relational information is naturally used is in a graph neural network for modelling object interactions and predicting the next time frame. We denote node representations by $\zeta_i^t$, edge representations by $\epsilon_{i,j}^t$, and actions applied to each node by $a_i^t$. Without DVNC, the changes of each nodes after each time step is computed by $\zeta_i^{t+1} = \zeta_i^t + \Delta\zeta_i^t$, where $\Delta\zeta_i^t = f_{node}(\zeta_i^t, a_t^i, \sum_{j \neq i} \epsilon_t^{i,j})$ and $\epsilon_{i,j}^t = f_{edge}(\zeta_i^t, z_j^t)$.

In this present work, we discretize the sum of all edges connected to each node with DVNC, as so:
$\Delta\zeta_i^t = f_{node}(\zeta_i^t, a_t^i, q(\sum_{j \neq i} \epsilon_t^{i,j}, L, G))$.

**Communication Between Positions in Transformers** In a transformer model without DVNC, at each layer, the scaled dot product multi-head soft attention is applied to allow the model to jointly attend to information from different representation subspaces at different positions (Vaswani et al., 2017) as:
$$\text{Output} = \text{residual} + \text{MULTIHEADATTENTION}(B, K, V),$$
where $\text{MULTIHEADATTENTION}(B, K, V) = \text{CONCATENATE}(head_1, head_2, .....head_n)W^O$ and $head_i = \text{SOFTATTENTION}(BW_i^B, KW_i^K, VW_i^V)$. Here, $W^O, W^B, W^K$, and $W^V$ are projection matrices, and $B$, $K$, and $V$ are queries, keys, and values respectively.

In this present work, we applied the DVNC process to the results of the attention in the last two layers of transformer model, as so:
$$\text{Output} = \text{residual} + q(\text{MULTIHEADATTENTION}(B, K, V), L, G).$$

**Communication with Modular Recurrent Neural Networks** There have been many efforts to introduce modularity into RNN. Recurrent independent mechanisms (RIMs) activated different

modules at different time step based on inputs (Goyal et al., 2019). In RIMs, outputs from different modules are communicated to each other via soft attention mechanism. In the original RIMs method, we have $\hat{z}_i^{t+1} = \text{RNN}(z_i^t, x^t)$ for active modules, and $\hat{z}_{i'}^{t+1} = z_{i'}^t$ for inactive modules, where $t$ is the time step, $i$ is index of the module, and $x_t$ is the input at time step $t$. Then, the dot product query-key soft attention is used to communication output from all modules $i \in \{1, \ldots, M\}$ such that $h_i^{t+1} = \text{SOFTATTENTION}(\hat{z}_1^{t+1}, \hat{z}_2^{t+1}, \ldots \hat{z}_M^{t+1})$.

In this present work, we applied the DVNC process to the output of the soft attention, like so: $z_i^{t+1} = \hat{z}_i^{t+1} + q(h_i^{t+1}, L, G)$. Appendix E presents the pseudocode for RIMs with discretization.

## 3 Related Works

**The Society of Specialists**    Our work is related to the theoretical nature of intelligence proposed by Minsky (1988) and others (Braitenberg, 1986; Fodor, 1983), in which the authors suggest that an intelligent mind can be built from many little specialist parts, each mindless by itself. Dividing model architecture into different specialists been the subject of a number of research directions, including neural module networks (Andreas et al., 2016)), multi-agent reinforcement learning (Zhang et al., 2019) and many others (Jacobs et al., 1991; Reed and De Freitas, 2015; Rosenbaum et al., 2019). Specialists for different computation processes have been introduced in many models such as RNNs and transformers (Goyal et al., 2019; Lamb et al., 2021; Goyal et al., 2021b). Specialists for entities or objects in fields including computer vision (Kipf et al., 2019). Methods have also been proposed to taking both entity and computational process into consideration (Goyal et al., 2020). In a more recent work, in addition to entities and computations, rules were considered when designing specialists (Goyal et al., 2021a). Our Discrete-Valued Neural Communication method can be seen as introducing specialists of representation into machine learning model.

**Communication among Specialists**    Efficient communication among different specialist components in a model requires compatible representations and synchronized messages. In recent years, attention mechanisms are widely used for selectively communication of information among specialist components in machine learning modes (Goyal et al., 2019, 2021b,a) and transformers (Vaswani et al., 2017; Lamb et al., 2021).collective memory and shared RNN parameters have also been used for multi-agent communication (Garland and Alterman, 1996; Pesce and Montana, 2020). Graph-based models haven been widely used in the context of relational reasoning, dynamical system simulation , multi-agent systems and many other fields. In graph neural networks, communication among different nodes were through edge attributes that are learned from the nodes the edge is connecting together with other information (Kipf et al., 2019; Scarselli et al., 2008; Bronstein et al., 2017; Watters et al., 2017; Van Steenkiste et al., 2018; Kipf et al., 2018; Battaglia et al., 2018; Tacchetti et al., 2018; Veerapaneni et al., 2019).Graph based method represent entities, relations, rules and other elements as node, edge and their attributes in a graph (Koller and Friedman, 2009; Battaglia et al., 2018). In graph architectures, the inductive bias is assuming the system to be learnt can be represented as a graph. In this study, our DVNC introduces inductive bias that forces inter-component communication to be discrete and share the same codebook. The combinatorial properties come from different combinations of latent vectors in each head and different combination of heads in each representation vector.While most of inter-specialist communication mechanisms operates in a pairwise symmetric manner, Goyal et al. (2021b) introduced a bandwidth limited communication channel to allow information from a limited number of modules to be broadcast globally to all modules, inspired by Global workspace theory (Baars, 2019, 1993).our proposed method selectively choose what information can communicated from each module. We argue these two methods are complimentary to each other and can be used together, which we like to investigate in future studies.

## 4 Experiments

In this study we have two hypothesis: 1) The use of discrete symbols limits the bandwidth of communication. 2) The use of shared discrete symbols will promote systematic generalization.Theoretical results obtained in section 2, agree with hypothesis 1. In order to verify hypothesis 2, in this section, we design and conduct empirical experiments to show that discretization of inter-component communication improves OOD generalization and model performance.

Table 2: Performance of Transformer Models with Discretized Communication on the Sort-of-Clevr Visual Reasoning Task.

| Method | Ternary Accuracy | Binary Accuracy | Unary Accuracy |
|---|---|---|---|
| Transformer baseline | $57.25 \pm 1.30$ | $76.00 \pm 1.41$ | $97.75 \pm 0.83$ |
| Discretized transformer (G=16) | $61.33 \pm 2.62$ | $84.00 \pm 2.94$ | $98.00 \pm 0.89$ |
| Discretized transformer (G=8) | $62.67 \pm 1.70$ | $88.00 \pm 0.82$ | $98.75 \pm 0.43$ |
| Discretized transformer (G=1) | $58.50 \pm 4.72$ | $80.50 \pm 7.53$ | $98.50 \pm 0.50$ |

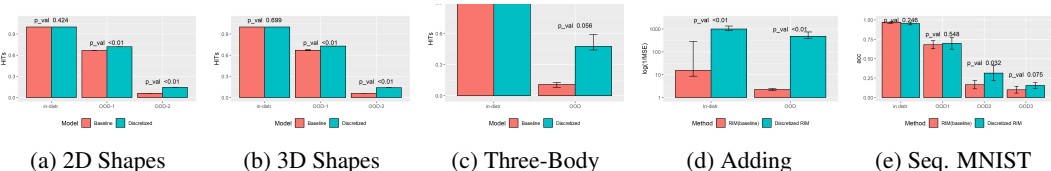

| (a) 2D Shapes | (b) 3D Shapes | (c) Three-Body | (d) Adding | (e) Seq. MNIST |
|---|---|---|---|---|

Figure 4: GNN models (a,b and c) and RIMs with DVNC (d and e) show improved OOD generalization. Test data in OOD 1 to 3 are increasingly different from training distribution.

## 4.1 Communication along Edges of Graph Neural Networks

We first turn our attention to visual reasoning task using GNNs and show that GNNs with DVNC have improved OOD generalization. In the tasks, the sequences of image frames in test set have different distribution from data used for training. We set up the model in a identical manner as in Kipf et al. 2019 (Kipf et al., 2019) except the introduction of DVNC. A CNN is used to to extract different objects from images. Objects are represented as nodes and relations between pairs of objects are represented as edges in the graph. The changes of nodes and edges can be learned by message passing in GNN. The information passed by all edges connected to a node is discretized by DVNC (see details in Section 2).

The following principles are followed when choosing and designing GNN visual reasoning tasks as well as all other OOD generalization tasks in subsequent sections: 1) Communication among different components are important for the tasks. 2) meaning of information communicated among components can be described in a discrete manner. 3) In OOD setting,distributions of information communicated among specialist components in test data also deviate from that in training data.

**Object Movement Prediction in Grid World** We begin with the tasks to predict object movement in grid world environments. Objects in the environments are interacting with each other and their movements are manipulated by actions applied to each object that give a push to the object in a randomly selected direction . Machine learning models are tasked to predict next positions of all objects in the next time step $t + 1$ given their positions and actions in time step $t$. We can think of this task as detecting whether an object can be pushed toward a direction. If object is blocked by the environment or another object in certain direction, then it can not be moved in that direction. Positions of objects are captures by nodes in GNN and the relative positions among different objects can be communicated in a discrete manner via message passing through edges.

We adapted and modified the original 2D shapes and 3D shapes movement tasks from Kipf et al. (2019) by introducing different number of objects in training or testing environment. In both 2D and 3D shapes tasks ,five objects are available in training data, three objects are available in OOD-1 and only two objects are available in OOD-2. Our experimental results suggest that DVNC in GNN improved OOD generalization in a statistically significant manner (Figure 4). In addition, the improvement is robust across different hyperparameters $G$ and $L$ (Figure 5). Details of the visual reasoning tasks set up and model hyperparameters can be found in Appendix.

**Three-body Physics Object Movement Prediction** Next, we turn our attention to a three-body-physics environment in which three balls interacting with each other and move according to physical laws in classic mechanics in a 2D environment. There are no external actions applied on the objects. We adapted and modified the three-body-physics environment from (Kipf et al., 2019). In OOD experiment, the sizes of the balls are different from the original training data. Details of the

Table 3: Graph Neural Networks benefit from discretized communication on OOD generalization in predicting movement in Atari games.

| Game | GNN (Baseline) | GNN (Discretized) | P-Value (vs. baseline) | Game | GNN (Baseline) | GNN (Discretized) | P-Value (vs. baseline) |
|---|---|---|---|---|---|---|---|
| Alien | $0.1991 \pm 0.0786$ | $0.2876 \pm 0.0782$ | 0.00019 | DoubleDunk | $0.8680 \pm 0.0281$ | $0.8793 \pm 0.0243$ | 0.04444 |
| BankHeist | $0.8224 \pm 0.0323$ | $0.8459 \pm 0.0490$ | 0.00002 | MsPacman | $0.2005 \pm 0.0362$ | $0.2325 \pm 0.0648$ | 0.05220 |
| Berzerk | $0.6077 \pm 0.0472$ | $0.6233 \pm 0.0509$ | 0.06628 | Pong | $0.1440 \pm 0.0845$ | $0.2965 \pm 0.1131$ | 0.00041 |
| Boxing | $0.9228 \pm 0.0806$ | $0.9502 \pm 0.0314$ | 0.69409 | SpaceInvaders | $0.0460 \pm 0.0225$ | $0.0820 \pm 0.0239$ | 0.00960 |

experimental set up can be found in Appendix. Our experimental results show that GNN with DVNC improved OOD generalization (Figure 4 ).

**Movement Prediction in Atari Games**    Similarly, we design OOD movement prediction tasks for 8 Atari games. Changes of each image frame depends on previous image frame and actions applied upon different objects. A different starting frame number is used to make the testing set OOD. GNN with DVNC showed statistically significant improvement in 6 out of 8 games and marginally significant improvement in the other games (Table 3).

## 4.2    Communication Between Positions in Transformers

In transformer models, attention mechanism is used to communicate information among different position. We design and conduct two visual reasoning tasks to understand if discretizing results of attention in transformer models will help improve the performance (Section 2). In the tasks, transformers take sequence of pixel as input and detect relations among different objects which are communicated through attention mechanisms.

We experimented with the Sort-of-CLEVR visual relational reasoning task, where the model is tasked with answering questions about certain properties of various objects and their relations with other objects (Santoro et al., 2017). Each image in Sort-of-CLEVR contains randomly placed geometrical shapes of different colors and shapes. Each image comes with 10 relational questions and 10 non-relational questions. Nonrelational questions only consider properties of individual objects. On the other hand, relational questions consider relations among multiple objects. The input to the model consists of the image and the corresponding question. Each image and question come with a finite number of possible answers and hence this task is to classify and pick the correct the answer(Goyal et al., 2021b). Transformer models with DVNC show significant improvement (Table 2).

## 4.3    Communication with Modular Recurrent Neural Networks

Recurrent Independent Mechanisms(RIMs) are RNN with modular structures. In RIMs, units communicate with each other using attention mechanisms at each time step. We discretize results of inter-unit attention in RIMs (Section 2). We conducted a numeric reasoning task and a visual reasoning task to understand if DVNC applied to RIMs improves OOD generalization.

We considered a synthetic adding task in which the model is trained to compute the sum of a sequence of numbers followed by certain number of dummy gap tokens(Goyal et al., 2019). In OOD settings of the task, the number of gap tokens after the target sequence is different in test set from training data. Our results show that DVNC makes significnat improvement in the OOD task(Figure 4).

For further evidence that RIMs with DVNC can better achieve OOD generlization, we consider the task of classifying MNIST digits as sequences of pixels (Krueger et al., 2016) and assay generalization to images of resolutions different from those seen during training. Our results suggest that RIMs with DVNC have moderately better OOD generalization than the baseline especially when the test images are very different from original training data (Figure 4).

## 4.4    Analysis and Ablations

**Discretizing Communication Results is Better than Discretizing other Parts of the Model:**
The intuition is that discretizing the results of communication with a shared codebook encourages more reliable and independent processing by different specialists. We experimentally tested this key hypothesis on the source of the DVNC's success by experimenting with discretizing other parts of the model. For example, in RIMs, we tried discretizing the updates to the recurrent state and

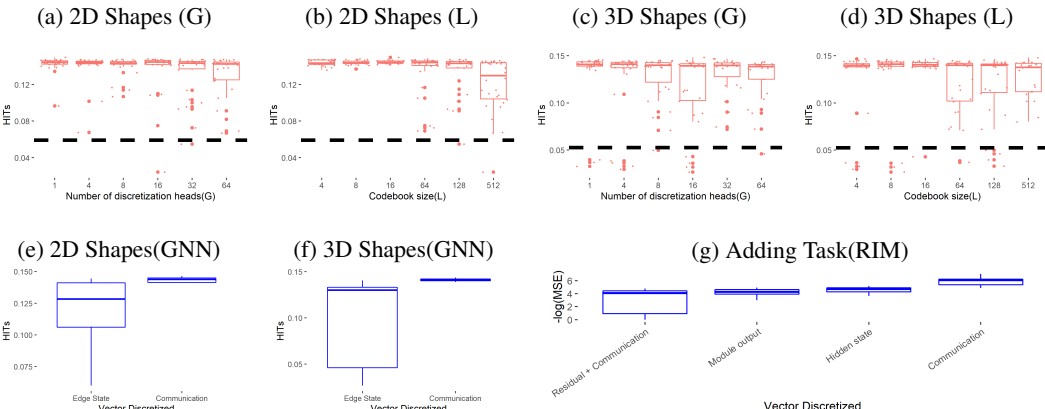

Figure 5: Upper Row: Models with DVNC have improved OOD generalization in wide range of hyperparameter settings. Red dots are performances of models with DVNC with different codebook size (L) and number of heads (G). Black dashed lines are performance of baseline methods without discretization. Lower row: (e) and (f) compare OOD generalization in HITs (higher is better) between GNNs with results of communication discretized vs. edge states discretized. (g) Compares RIMs model with results of communication discretized vs. other vectors discretized.

tried discretizing the inputs. On the adding task, this led to improved results over the baseline, but performed much worse than discretizing communication. For GNNs we tried discretizing the input to the communication (the edge hidden states) instead of the result of communication, and found that it led to significantly worse results and had very high variance between different trials. These results are in Figure 5.

**VQ-VAE Discretization Outperforms Gumbel-Softmax:**   The main goal of our work was to demonstrate the benefits of communication with discrete values, and this discretization could potentially be done through a number of different mechanisms. Experimentally, we found that the nearest-neighbor and straight-through estimator based discretization technique,similar to method used in VQ-VAE, outperformed the use of a Gumbel-Softmax to select the discrete tokens (Figure 6 in Appendix). These positive results led us to focus on the VQ-VAE discretization technique, but in principle other mechanisms could also be recruited to accomplish our discrete-valued neural communication framework. We envision that DVCN should work complementarily with future advances in learning discrete representations.

## 5   Discussion

With the evolution of deep architectures from the monolithic MLP to complex architectures with specialized components, we are faced with the issue of how to ensure that subsystems can communicate and coordinate. Communication via continuous, high-dimensional signals is a natural choice given the history of deep learning, but our work argues that discretized communication results in more robust, generalizable learning. Discrete-Valued Neural Communication (DVNC) achieves a much lower noise-sensitivity bound while allowing high expressiveness through the use of multiple discretization heads. This technique is simple and easy-to-use in practice and improves out-of-distribution generalization. DVNC is applicable to all structured architectures that we have examined where inter-component communication is important and the information to be communicated can be discretized by its nature.

**Limitations**   The proposed method has two major limitations. First, DVNC can only improve performance if communication among specialists is important for the task. If the different components do not have good specialization, then DVNC's motivation is less applicable. Another limitation is that the discretization process can reduce the expressivity of the function class, although this can be mitigated by using a large value for $G$ and $L$ and can be partially monitored by the quantity of training data (e.g., training loss) similarly to the principle of structural minimization. Hence future work could examine how to combine discrete communication and continuous communication.

**Social Impact** Research conducted in this study is purely technical. The authors expect no direct negative nor positive social impact.

**Funding in direct support of this work** Funding from Samsung Electronics Ltd

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
