*Supplementary material for*

# Discrete-Valued Neural Communication
# in Structured Architectures Enhances Generalization

Dianbo Liu, Alex Lamb, Kenji Kawaguchi,
Anirudh Goyal, Chen Sun, Michael C. Mozer, & Yoshua Bengio

## A  Additional theorems for theoretical motivations

In this appendix, as a complementary to Theorems 1–2, we provide additional theorems, Theorems 3–4, which further illustrate the two advantages of the discretization process by considering an abstract model with the discretization bottleneck. For the advantage on the sensitivity, the error due to potential noise and perturbation without discretization — the third term $\xi(w, r', \mathcal{M}', d) > 0$ in Theorem 4 — is shown to be minimized to zero with discretization in Theorems 3. For the second advantage, the underlying dimensionality of $\mathcal{N}_{(\mathcal{M}', d')}(r', \mathcal{H}) + \ln(\mathcal{N}_{(\mathcal{M}, d)}(r, \Theta)/\delta)$ without discretization (in the bound of Theorem 4) is proven to be reduced to the typically much smaller underlying dimensionality of $L^G + \ln(\mathcal{N}_{(\mathcal{M}, d)}(r, E \times \Theta))$ with discretization in Theorems 3. Here, for any metric space $(\mathcal{M}, d)$ and subset $M \subseteq \mathcal{M}$, the $r$-converging number of $M$ is defined by $\mathcal{N}_{(\mathcal{M}, d)}(r, M) = \min\left\{|\mathcal{C}| : \mathcal{C} \subseteq \mathcal{M}, M \subseteq \cup_{c \in \mathcal{C}} \mathcal{B}_{(\mathcal{M}, d)}[c, r]\right\}$ where the (closed) ball of radius $r$ at centered at $c$ is denoted by $\mathcal{B}_{(\mathcal{M}, d)}[c, r] = \{x \in \mathcal{M} : d(x, c) \le r\}$. See Appendix C.1 for a simple comparison between the bound of Theorem 3 and that of Theorem 4 when the metric spaces $(\mathcal{M}, d)$ and $(\mathcal{M}', d')$ are chosen to be Euclidean spaces.

We now introduce the notation used in Theorems 3–4. Let $q_e(h) := q(h, L, G)$. The models are defined by $\tilde{f}(x) := \tilde{f}(x, w, \theta) := (\varphi_w \circ h_\theta)(x)$ without the discretization and $f(x) := f(x, w, e, \theta) := (\varphi_w \circ q_e \circ h_\theta)(x)$ with the discretization. Here, $\varphi_w$ represents a deep neural network with weight parameters $w \in \mathcal{W} \subset \mathbb{R}^D$, $q_e$ is the discretization process with the codebook $e \in E \subset \mathbb{R}^{L \times m}$, and $h_\theta$ represents a deep neural network with parameters $\theta \in \Theta \subset \mathbb{R}^\zeta$. Thus, the tuple of all learnable parameters are $(w, e, \theta)$. For the codebook space, $E = E_1 \times E_2$ with $E_1 \subset \mathbb{R}^L$ and $E_2 \subset \mathbb{R}^m$. Moreover, let $J : (f(x), y) \mapsto J(f(x), y) \in \mathbb{R}$ be an arbitrary (fixed) function, $h_\theta(x) \in \mathcal{H} \subset \mathbb{R}^m$, $x \in \mathcal{X}$, and $y \in \mathcal{Y} = \{y^{(1)}, y^{(2)}\}$ for some $y^{(1)}$ and $y^{(2)}$.

**Theorem 3.** (with discretization) *Let $C_J(w)$ be the smallest real number such that $|J(\varphi_w(\eta), y)| \le C_J(w)$ for all $(\eta, y) \in E_2 \times \mathcal{Y}$. Let $\rho \in \mathbb{N}^+$ and $(\mathcal{M}, d)$ be a matric space such that $E \times \Theta \subseteq \mathcal{M}$. Then, for any $\delta > 0$, with probability at least $1 - \delta$ over an iid draw of $n$ examples $((x_i, y_i))_{i=1}^n$, the following holds: for any $(w, e, \theta) \in \mathcal{W} \times E \times \Theta$,*

$$\left| \mathbb{E}_{x,y}[J(f(x, w, e, \theta), y)] - \frac{1}{n} \sum_{i=1}^n J(f(x_i, w, e, \theta), y_i) \right|$$
$$\le C_J(w) \sqrt{\frac{4L^G \ln 2 + 2 \ln(\mathcal{N}_{(\mathcal{M}, d)}(r, E \times \Theta)/\delta)}{n}} + \sqrt{\frac{\mathcal{L}_d(w)^{2/\rho}}{n}},$$

*where $r = \mathcal{L}_d(w)^{1/\rho - 1} \sqrt{\frac{1}{n}}$ and $\mathcal{L}_d(w) \ge 0$ is the smallest real number such that for all $(e, \theta)$ and $(e', \theta')$ in $E \times \Theta$, $|\psi_w(e, \theta) - \psi_w(e', \theta')| \le \mathcal{L}_d(w) d((e, \theta), (e', \theta'))$ with $\psi_w(e, \theta) = \mathbb{E}_{x,y}[J(f(x), y)] - \frac{1}{n} \sum_{i=1}^n J(f(x_i), y_i)$*

**Theorem 4.** (without discretization) *Let $\tilde{C}_J(w)$ be the smallest real number such that $|J((\varphi_w \circ h_\theta)(x), y)| \le \tilde{C}_J(w)$ for all $(\theta, x, y) \in \Theta \times \mathcal{X} \times \mathcal{Y}$. Let $\rho \in \mathbb{N}^+$ and $(\mathcal{M}, d)$ be a matric space such that $\Theta \subseteq \mathcal{M}$. Let $(\mathcal{M}', d')$ be a matric space such that $\mathcal{H} \subseteq \mathcal{M}'$. Fix $r' > 0$ and $\bar{\mathcal{C}}_{r', d'} \in \arg\min_{\mathcal{C}} \{|\mathcal{C}| : \mathcal{C} \subseteq \mathcal{M}', \mathcal{H} \subseteq \cup_{c \in \mathcal{C}} \mathcal{B}_{(\mathcal{M}', d')}[c, r']\}$. Assume that for any $c \in \bar{\mathcal{C}}_{r', d'}$, we have $|(J(\varphi_w(h), y) - (J(\varphi_w(h'), y)| \le \xi(w, r', \mathcal{M}', d)$ for any $h, h' \in \mathcal{B}_{(\mathcal{M}', d')}[c, r']$ and $y \in \mathcal{Y}$. Then, for any $\delta > 0$, with probability at least $1 - \delta$ over an iid draw of $n$ examples $((x_i, y_i))_{i=1}^n$, the following holds: for any $(w, \theta) \in \mathcal{W} \times \Theta$,*

$$\left| \mathbb{E}_{x,y}[J(\tilde{f}(x, w, \theta), y)] - \frac{1}{n} \sum_{i=1}^n J(\tilde{f}(x_i, w, \theta), y_i) \right|$$

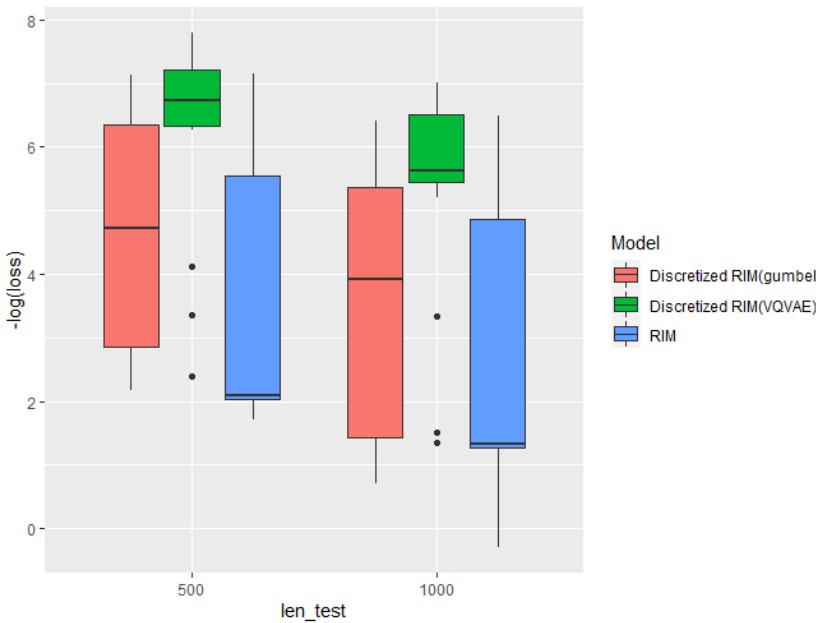

Figure 6: Performance on adding task (RIMs) with no discretization, Gumbel-Softmax discretization, or VQ-VAE style discretization (ours). Test length=500 is in-distribution test result and test length=1000 is out-of-distribution results.

$$\leq \tilde{C}_J(w)\sqrt{\frac{4\mathcal{N}_{(\mathcal{M}',d')}(r',\mathcal{H})\ln 2 + 2\ln(\mathcal{N}_{(\mathcal{M},d)}(r,\Theta)/\delta)}{n}} + \sqrt{\frac{\tilde{\mathcal{L}}_d(w)^{2/\rho}}{n}} + \xi(w,r',\mathcal{M}',d),$$

where $r = \tilde{\mathcal{L}}_d(w)^{1/\rho-1}\sqrt{\frac{1}{n}}$ and $\tilde{\mathcal{L}}_d(w) \geq 0$ is the smallest real number such that for all $\theta$ and $\theta'$ in $\Theta$, $|\tilde{\psi}_w(\theta) - \tilde{\psi}_w(\theta')| \leq \tilde{\mathcal{L}}_d(w)d(\theta,\theta')$ with $\tilde{\psi}_w(\theta) = \mathbb{E}_{x,y}[J(\tilde{f}(x),y)] - \frac{1}{n}\sum_{i=1}^n J(\tilde{f}(x_i),y_i)$.

Note that we have $C_J(w) \leq \tilde{C}_J(w)$ and $\mathcal{L}_d(w) \approx \tilde{\mathcal{L}}_d(w)$ by their definition. For example, if we set $J$ to be a loss criterion, the bound in Theorem 4 becomes in the same order as and comparable to the generalization bound via the *algorithmic robustness* approach proposed by the previous papers (Xu and Mannor, 2012; Sokolic et al., 2017a,b), as we show in Appendix C.2.

# B  Additional Experiments

# C  Additional discussions on theoretical motivations

## C.1  Simple comparison of Theorems 3 and 4 with Euclidean space

For the purpose of of the comparison, we will now consider the simple worst case with no additional structure with the Euclidean space to instantiate $\mathcal{N}_{(\mathcal{M}',d')}(r',\mathcal{H})$, $\mathcal{N}_{(\mathcal{M},d)}(r,\Theta)$, and $\mathcal{N}_{(\mathcal{M},d)}(r, E \times \Theta)$. It should be obvious that we can improve the bounds via considering metric spaces with additional structures. For example, we can consider a lower dimensional manifold $\mathcal{H}$ in the ambient space of $\mathbb{R}^m$ to reduce $\mathcal{N}_{(\mathcal{M}',d')}(r',\mathcal{H})$. Similar ideas can be applied for $\Theta$ and $E \times \Theta$. Furthermore, the invariance as well as margin were used to reduce the bound on $\mathcal{N}_{(\mathcal{M}',d')}(r',\mathcal{X})$ in previous works (Sokolic et al., 2017a,b) and similar ideas can be applied for $\mathcal{N}_{(\mathcal{M}',d')}(r',\mathcal{H})$, $\mathcal{N}_{(\mathcal{M},d)}(r,\Theta)$, and $\mathcal{N}_{(\mathcal{M},d)}(r, E \times \Theta)$. In this regard, the discretization can be viewed as a method to minimize $\mathcal{N}_{(\mathcal{M}',d')}(r',\mathcal{H})$ to easily controllable $L^G$ while minimizing the sensitivity term $\xi(w,r',\mathcal{M}',d)$ to zero at the same time in Theorems 3 and 4.

Suppose that for any $y \in \mathcal{Y}$, the function $h \mapsto J(\varphi_w(h), y)$ is Lipschitz continuous as $|(J(\varphi_w(h), y) - (J(\varphi_w(h'), y)| \leq \varsigma(w) d(h, h')$. Then, we can set $\xi(w, r', \mathcal{M}', d) = 2\varsigma(w) r'$ since $d(h, h') \leq 2r'$ for any $h, h' \in \mathcal{B}_{(\mathcal{M}', d')}[c, r']$.

As an simple example, let us choose the metric space $(\mathcal{M}', d')$ to be the Euclidean space $\mathbb{R}^m$ with the Euclidean metric and $\mathcal{H} \subset \mathbb{R}^m$ such that $\|v\|_2 \leq R_{\mathcal{H}}$ for all $v \in \mathcal{H}$. Then, we have $\mathcal{N}_{(\mathcal{M}', d')}(r', \mathcal{H}) \leq (2R_{\mathcal{H}} \sqrt{m}/r')^m$ and we can set $\xi(w, r', \mathcal{M}', d) = 2\varsigma(w) r'$. Thus, by setting $r' = R_{\mathcal{H}}/2$, we can replace $\mathcal{N}_{(\mathcal{M}', d')}(r', \mathcal{H})$ by $(4\sqrt{m})^m$ and set $\xi(w, r', \mathcal{M}', d) = \varsigma(w) R_{\mathcal{H}}$.

Similarly, let us choose the metric space $(\mathcal{M}, d)$ to be the Euclidean space with the Euclidean metric and $E \subset \mathbb{R}^{Lm}$ and $\Theta \subset \mathbb{R}^{\zeta}$ such that $\|v\|_2 \leq R_E$ for all $v \in E$ and $\|v\|_2 \leq R_{\Theta}$ for all $v \in \Theta$. This implies that $\|(v_E, v_\theta)\|_2 \leq \sqrt{R_E^2 + R_{\Theta}^2}$. Thus, we have $\mathcal{N}_{(\mathcal{M}, d)}(r, \Theta) \leq (2R_{\Theta} \sqrt{\zeta}/r)^{\zeta}$ and $\mathcal{N}_{(\mathcal{M}, d)}(r, E \times \Theta) \leq (2\sqrt{R_E^2 + R_{\Theta}^2} \sqrt{Lm + \zeta}/r)^{Lm+\zeta}$. Since $r = \tilde{\mathcal{L}}_d(w)^{1/\rho - 1} \sqrt{\frac{1}{n}}$ and $r = \mathcal{L}_d(w)^{1/\rho - 1} \sqrt{\frac{1}{n}}$, we can replace $\mathcal{N}_{(\mathcal{M}, d)}(r, \Theta)$ by $(2R_{\Theta} \tilde{\mathcal{L}}_d(w)^{1-1/\rho} \sqrt{\zeta n})^{\zeta}$ and $\mathcal{N}_{(\mathcal{M}, d)}(r, E \times \Theta)$ by $(2\mathcal{L}_d(w)^{1-1/\rho} \sqrt{R_E^2 + R_{\Theta}^2} \sqrt{(Lm + \zeta)n})^{Lm+\zeta}$. By summarizing these and ignoring the logarithmic dependency as in the standard $\tilde{O}$ notation, we have the following bounds for Theorems 3 and 4:

$$\text{(with discretization)} \quad C_J(w) \sqrt{\frac{4L^G + 2Lm + 2\zeta + 2\ln(1/\delta)}{n}} + \sqrt{\frac{\mathcal{L}_d(w)^{2/\rho}}{n}},$$

and

$$\text{(without discretization)} \quad \tilde{C}_J(w) \sqrt{\frac{4(4\sqrt{m})^m + 2\zeta + 2\ln(1/\delta)}{n}} + \sqrt{\frac{\tilde{\mathcal{L}}_d(w)^{2/\rho}}{n}} + \varsigma(w) R_{\mathcal{H}},$$

where we used the fact that $\ln(x/y) = \ln(x) + \ln(1/y)$. Here, we can more easily see that the discretization process has the benefits in the two aspects:

1. The discretization process improves sensitivity against noise and perturbations: i.e., it reduces the sensitivity term $\varsigma(w) R_{\mathcal{H}}$ to be zero.

2. The discretization process reduces underlying dimensionality: i.e., it reduce the term of $4(4\sqrt{m})^m$ to the term of $4L^G + 2Lm$. In practice, we typically have $4(4\sqrt{m})^m \gg 4L^G + 2Lm$. This shows that using the discretization process withe codebook of size $L \times m$, we can successfully reduce the exponential dependency on $m$ to the linear dependency on $m$. This is a significant improvement.

## C.2 On the comparison of Theorem 4 and algorithmic robustness

If we assume that the function $x \mapsto \ell(\tilde{f}(x), y)$ is Lipschitz for all $y \in \mathcal{Y}$ with Lipschitz constant $\varsigma_x(w)$ similarly to our assumption in Theorem 4, the bound via the algorithmic robustness in the previous paper (Xu and Mannor, 2012) becomes the following: for any $\delta > 0$, with probability at least $1 - \delta$ over an iid draw of $n$ examples $((x_i, y_i))_{i=1}^n$, for any $(w, \theta) \in \mathcal{W} \times \Theta$,

$$\left| \mathbb{E}_{x,y}[\ell(\tilde{f}(x, w, \theta), y)] - \frac{1}{n} \sum_{i=1}^n [\ell(\tilde{f}(x_i, w, \theta), y_i)] \right| \tag{4}$$

$$\leq \hat{C}_J \sqrt{\frac{4\mathcal{N}_{(\mathcal{M}', d')}(r', \mathcal{X}) \ln 2 + 2\ln \frac{1}{\delta}}{n}} + 2\varsigma_x(w) r',$$

where $\hat{C}_J \geq \tilde{C}_J(w)$ for all $w \in \mathcal{W}$ and $(\mathcal{M}', d')$ is a metric space such that $\mathcal{X} \subseteq \mathcal{M}'$. See Appendix C.3. for more details on the algorithmic robustness bounds.

Thus, we can see that the dominant term $\mathcal{N}_{(\mathcal{M}', d')}(r', \mathcal{H})$ in Theorem 4 is comparable to the dominant term $\mathcal{N}_{(\mathcal{M}', d')}(r', \mathcal{X})$ in the previous study. Whereas the previous bound measures the robustness in the input space $\mathcal{X}$, the bound in Theorem 4 measures the robustness in the bottleneck layer space $\mathcal{H}$. When compared to the input space $\mathcal{X}$, if the bottleneck layer space $\mathcal{H}$ is smaller or has more structures, then we can have $\mathcal{N}_{(\mathcal{M}', d')}(r', \mathcal{H}) < \mathcal{N}_{(\mathcal{M}', d')}(r', \mathcal{X})$ and Theorem 4 can be advantageous over the previous bound. However, Theorem 4 is not our main result as we have much tighter bounds for the discretization process in Theorem 3 as well as Theorem 1.

## C.3 On algorithmic robustness

In the previous paper, algorithmic robustness is defined to be the measure of how much the loss value can vary with respect to the perturbations of values data points $(x, y) \in \mathcal{X} \times \mathcal{Y}$. More precisely, an algorithm $\mathcal{A}$ is said to be $(|\Omega|, \varrho(\cdot))$-robust if $\mathcal{X} \times \mathcal{Y}$ can be partitioned into $|\Omega|$ disjoint sets $\Omega_1, \ldots, \Omega_{|\Omega|}$ such that for any dataset $S \in (\mathcal{X} \times \mathcal{Y})^m$, all $(x, y) \in S$, all $(x', y') \in \mathcal{X} \times \mathcal{Y}$, and all $i \in \{1, \ldots, |\Omega|\}$, if $(x, y), (x', y') \in \Omega_i$, then

$$|\ell(\tilde{f}(x), y) - \ell(\tilde{f}(x'), y')| \leq \varrho(S).$$

If algorithm $\mathcal{A}$ is $(\Omega, \varrho(\cdot))$-robust and the codomain of $\ell$ is upper-bounded by $M$, then given a dataset $S$, we have (Xu and Mannor, 2012) that for any $\delta > 0$, with probability at least $1 - \delta$,

$$\left| \mathbb{E}_{x,y}[\ell(\tilde{f}(x), y)] - \frac{1}{n} \sum_{i=1}^{n} [\ell(\tilde{f}(x_i), y_i)] \right| \leq M \sqrt{\frac{2|\Omega| \ln 2 + 2 \ln \frac{1}{\delta}}{n}} + \varrho(S).$$

The previous paper (Xu and Mannor, 2012) further shows concrete examples of this bound for a case where the function $(x, y) \mapsto \ell(\tilde{f}(x), y)$ is Lipschitz with Lipschitz constant $\varsigma_{x,y}(w)$,

$$\left| \mathbb{E}_{x,y}[\ell(\tilde{f}(x), y)] - \frac{1}{n} \sum_{i=1}^{n} [\ell(\tilde{f}(x_i), y_i)] \right| \leq M \sqrt{\frac{2 \mathcal{N}_{(\mathcal{M}', d')}(r', \mathcal{X} \times \mathcal{Y}) \ln 2 + 2 \ln \frac{1}{\delta}}{n}} + 2 \varsigma_{x,y}(w) r',$$

where $(\mathcal{M}', d')$ is a metric space such that $\mathcal{X} \times \mathcal{Y} \subseteq \mathcal{M}'$. Note that the Lipschitz assumption on t he function $(x, y) \mapsto \ell(\tilde{f}(x), y)$ does not typically hold for the 0-1 loss on classification. For classification, we can assume that the function $x \mapsto \ell(\tilde{f}(x), y)$ is Lipschitz instead, yielding equation (4).

# D Proofs

We use the notation of $q_e(h) := q(h, L, G)$ in the proofs.

## D.1 Proof of Theorem 1

*Proof of Theorem 1.* Let $\mathcal{I}_k = \{i \in [n] : q_e(h_i) = Q_k\}$. By using the following equality,

$$\mathbb{E}_h[\phi_k^S(q_e(h))] = \mathbb{E}_h[\phi_k^S(q_e(h)) | q_e(h) = Q_k] \Pr(q_e(h) = Q_k) = \phi(Q_k) \Pr(q_e(h) = Q_k),$$

we first decompose the difference into two terms as

$$\mathbb{E}_h[\phi_k^S(q_e(h))] - \frac{1}{n} \sum_{i=1}^{n} \phi_k^S(q_e(h_i)) \tag{5}$$

$$= \phi(Q_k) \left( \Pr(q_e(h) = Q_k) - \frac{|\mathcal{I}_k|}{n} \right) + \left( \phi(Q_k) \frac{|\mathcal{I}_k|}{n} - \frac{1}{n} \sum_{i=1}^{n} \phi_k^S(q_e(h_i)) \right).$$

The second term in the right-hand side of (5) is further simplified by using

$$\frac{1}{n} \sum_{i=1}^{n} \phi_k^S(q_e(h_i)) = \frac{1}{n} \sum_{i \in \mathcal{I}_k} \phi(q_e(h_i)),$$

and

$$\phi(Q_k) \frac{|\mathcal{I}_k|}{n} = \frac{1}{n} \sum_{i \in \mathcal{I}_k} \phi(q_e(h_i)),$$

as

$$\phi(Q_k) \frac{|\mathcal{I}_k|}{n} - \frac{1}{n} \sum_{i=1}^{n} \phi_k^S(q_e(h_i)) = 0.$$

Substituting these into equation (5) yields

$$\left| \mathbb{E}_h[\phi_k^S(q_e(h))] - \frac{1}{n}\sum_{i=1}^n \phi_k^S(q_e(h_i)) \right| = \left| \phi(Q_k)\left( \Pr(q_e(h) = Q_k) - \frac{|\mathcal{I}_k|}{n} \right) \right|$$

$$\leq |\phi(Q_k)| \left| \Pr(q_e(h) = Q_k) - \frac{|\mathcal{I}_k|}{n} \right|. \qquad (6)$$

Let $p_k = \Pr(q_e(h) = Q_k)$ and $\hat{p} = \frac{|\mathcal{I}_k|}{n}$. Consider the random variable $X_i = \mathbb{1}\{q_e(h_i) = Q_k\}$ with the pushforward measure of the random variable $h_i$ under the map $h_i \mapsto \mathbb{1}\{q_e(h_i) = Q_k\}$. Here, we have that $X_i \in \{0, 1\} \subset [0, 1]$. Since $e$ is fixed and $h_1, \ldots, h_n$ are assumed to be iid, the Hoeffding's inequality implies the following: for each fixed $k \in [L^G]$,

$$\Pr(|p_k - \hat{p}_k| \geq t) \leq 2\exp\left( -2nt^2 \right).$$

By solving $\delta' = 2\exp\left( -2nt^2 \right)$, this implies that for each fixed $k \in [L^G]$, for any $\delta' > 0$, with probability at least $1 - \delta'$,

$$|p_k - \hat{p}_k| \leq \sqrt{\frac{\ln(2/\delta')}{2n}}.$$

By taking union bounds over $k \in [L^G]$ with $\delta' = \frac{\delta}{L^G}$, we have that for any $\delta > 0$, with probability at least $1 - \delta$, the following holds for all $k \in [L^G]$:

$$|p_k - \hat{p}_k| \leq \sqrt{\frac{\ln(2L^G/\delta)}{2n}}.$$

Substituting this into equation (6) yields that for any $\delta > 0$, with probability at least $1 - \delta$, the following holds for all $k \in [L^G]$:

$$\left| \mathbb{E}_h[\phi_k^S(q_e(h))] - \frac{1}{n}\sum_{i=1}^n \phi_k^S(q_e(h_i)) \right| \leq |\phi(Q_k)|\sqrt{\frac{\ln(2L^G/\delta)}{2n}} = |\phi(Q_k)|\sqrt{\frac{G\ln(L) + \ln(2/\delta)}{2n}}.$$

$\square$

### D.2 Proof of Theorem 2

*Proof of Theorem 2.* Let $(\mathcal{M}', d')$ be a matric space such that $\mathcal{H} \subseteq \mathcal{M}'$. Fix $r' > 0$ and $\bar{\mathcal{C}} \in \operatorname{argmin}_{\mathcal{C}}\{|\mathcal{C}| : \mathcal{C} \subseteq \mathcal{M}', \mathcal{H} \subseteq \cup_{c \in \mathcal{C}} \mathcal{B}_{(\mathcal{M}',d')}[c, r']\}$ such that $|\bar{\mathcal{C}}| < \infty$. Fix an arbitrary ordering and define $c_k \in \bar{\mathcal{C}}_{r',d'}$ to be the $k$-the element in the ordered version of $\bar{\mathcal{C}}$ in that fixed ordering (i.e., $\cup_k\{c_k\} = \bar{\mathcal{C}}_{r',d'}$). Let $\mathcal{B}[c] = \mathcal{B}_{(\mathcal{M}',d')}[c, r']$ and $S = \{\mathcal{B}[c_1], \mathcal{B}[c_2], \ldots, \mathcal{B}[c_{|\bar{\mathcal{C}}|}]\}$. Suppose that $\left| \phi_k^S(h) - \phi_k^S(h') \right| \leq \xi_k(r', \mathcal{M}', d)$ for any $h, h' \in \mathcal{B}[c_k]$ and $k \in [|\bar{\mathcal{C}}|]$, which is shown to be satisfied later in this proof. Let $\mathcal{I}_k = \{i \in [n] : h_i \in \mathcal{B}[c_k]\}$ for all $k \in [|\bar{\mathcal{C}}|]$. By using the following equality,

$$\mathbb{E}_h[\phi_k^S(h)] = \mathbb{E}_h[\phi_k^S(h)|h \in \mathcal{B}[c_k]]\Pr(h \in \mathcal{B}[c_k]),$$

we first decompose the difference into two terms as

$$\left| \mathbb{E}_h[\phi_k^S(h)] - \frac{1}{n}\sum_{i=1}^n \phi_k^S(h_i) \right| \qquad (7)$$

$$\leq \left| \mathbb{E}_h[\phi_k^S(h)|h \in \mathcal{B}[c_k]]\left( \Pr(h \in \mathcal{B}[c_k]) - \frac{|\mathcal{I}_k|}{n} \right) \right| + \left| \mathbb{E}_h[\phi_k^S(h)|h \in \mathcal{B}[c_k]]\frac{|\mathcal{I}_k|}{n} - \frac{1}{n}\sum_{i=1}^n \phi_k^S(h_i) \right|$$

The second term in the right-hand side of (7) is further simplified by using

$$\frac{1}{n}\sum_{i=1}^n \phi_k^S(h_i) = \frac{1}{n}\sum_{i \in \mathcal{I}_k} \phi_k^S(h_i),$$

and

$$\mathbb{E}_h[\phi_k^S(h)|h \in \mathcal{B}[c_k]]\frac{|\mathcal{I}_k|}{n} = \frac{1}{n}\sum_{i \in \mathcal{I}_k} \mathbb{E}_h[\phi_k^S(h)|h \in \mathcal{B}[c_k]],$$

as

$$\left| \mathbb{E}_h[\phi_k^S(h)|h \in \mathcal{B}[c_k]] \frac{|\mathcal{I}_k|}{n} - \frac{1}{n} \sum_{i=1}^n \phi_k^S(h_i) \right|$$

$$= \left| \frac{1}{n} \sum_{i \in \mathcal{I}_k} \left( \mathbb{E}_h[\phi_k^S(h)|h \in \mathcal{B}[c_k]] - \phi_k^S(h_i) \right) \right|$$

$$\leq \frac{1}{n} \sum_{i \in \mathcal{I}_k} \sup_{h \in \mathcal{B}[c_k]} \left| \phi_k^S(h) - \phi_k^S(h_i) \right| \leq \frac{|\mathcal{I}_k|}{n} \xi_k(r', \mathcal{M}', d).$$

Substituting these into equation (7) yields

$$\left| \mathbb{E}_h[\phi_k^S(h)] - \frac{1}{n} \sum_{i=1}^n \phi_k^S(h_i) \right|$$

$$\leq \left| \mathbb{E}_h[\phi_k^S(h)|h \in \mathcal{B}[c_k]] \left( \Pr(h \in \mathcal{B}[c_k]) - \frac{|\mathcal{I}_k|}{n} \right) \right| + \frac{|\mathcal{I}_k|}{n} \xi_k(r', \mathcal{M}', d)$$

$$\leq |\mathbb{E}_h[\phi(h)|h \in \mathcal{B}[c_k]]| \left| \left( \Pr(h \in \mathcal{B}[c_k]) - \frac{|\mathcal{I}_k|}{n} \right) \right| + \frac{|\mathcal{I}_k|}{n} \xi_k(r', \mathcal{M}', d), \tag{8}$$

Let $p_k = \Pr(h \in \mathcal{B}[c_k])$ and $\hat{p} = \frac{|\mathcal{I}_k|}{n}$. Consider the random variable $X_i = \mathbb{1}\{h \in \mathcal{B}[c_k]\}$ with the pushforward measure of the random variable $h_i$ under the map $h_i \mapsto \mathbb{1}\{h \in \mathcal{B}[c_k]\}$. Here, we have that $X_i \in \{0, 1\} \subset [0, 1]$. Since $\mathcal{B}[c_k]$ is fixed and $h_1, \ldots, h_n$ are assumed to be iid, the Hoeffding's inequality implies the following: for each fixed $k \in [|\bar{\mathcal{C}}|]$,

$$\Pr(|p_k - \hat{p}_k| \geq t) \leq 2 \exp\left(-2nt^2\right).$$

By solving $\delta' = 2 \exp\left(-2nt^2\right)$, this implies that for each fixed $k \in [|\bar{\mathcal{C}}|]$, for any $\delta' > 0$, with probability at least $1 - \delta'$,

$$|p_k - \hat{p}_k| \leq \sqrt{\frac{\ln(2/\delta')}{2n}}.$$

By taking union bounds over $k \in [|\bar{\mathcal{C}}|]$ with $\delta' = \frac{\delta}{|\bar{\mathcal{C}}|}$, we have that for any $\delta > 0$, with probability at least $1 - \delta$, the following holds for all $k \in [|\bar{\mathcal{C}}|]$:

$$|p_k - \hat{p}_k| \leq \sqrt{\frac{\ln(2|\bar{\mathcal{C}}|/\delta)}{2n}} = \sqrt{\frac{\ln(|\bar{\mathcal{C}}|) + \ln(2/\delta)}{2n}}.$$

Substituting this into equation (8) yields that for any $\delta > 0$, with probability at least $1 - \delta$, the following holds for all $k \in [|\bar{\mathcal{C}}|]$:

$$\left| \mathbb{E}_h[\phi_k^S(h)] - \frac{1}{n} \sum_{i=1}^n \phi_k^S(h_i) \right|$$

$$\leq |\mathbb{E}_h[\phi(h)|h \in \mathcal{B}[c_k]]| \sqrt{\frac{\ln(\mathcal{N}_{(\mathcal{M}',d')}(r', \mathcal{H})) + \ln(2/\delta)}{2n}}$$

$$+ \xi_k(r', \mathcal{M}', d) \left( \frac{1}{n} \sum_{i=1}^n \mathbb{1}\{h_i \in \mathcal{B}[c_k]\} \right),$$

where we used $|\mathcal{I}_k| = \sum_{i=1}^n \mathbb{1}\{h_i \in \mathcal{B}[c_k]\}$. Let us now choose the metric space $(\mathcal{M}', d')$ to be the Euclidean space $\mathbb{R}^m$ with the Euclidean metric and $\mathcal{H} \subset \mathbb{R}^m$ such that $\|v\|_2 \leq R_{\mathcal{H}}$ for all $v \in \mathcal{H}$. Then, we have $\mathcal{N}_{(\mathcal{M}',d')}(r', \mathcal{H}) \leq (2R_{\mathcal{H}}\sqrt{m}/r')^m$ and we can set $\xi(w, r', \mathcal{M}', d) = 2\varsigma_k r'$. This is because that the function $h \mapsto \phi_k^S(h)$ is Lipschitz continuous as $|\phi_k^S(h) - \phi_k^S(h')| \leq \varsigma_k d(h, h')$, and because $d(h, h') \leq 2r'$ for any $h, h' \in \mathcal{B}_{(\mathcal{M}',d')}[c_k, r']$. Thus, by setting $r' = R_{\mathcal{H}}/(2\sqrt{n})$, we can replace $\mathcal{N}_{(\mathcal{M}',d')}(r', \mathcal{H})$ by $(4\sqrt{nm})^m$ and set $\xi(w, r', \mathcal{M}', d) = \varsigma_k R_{\mathcal{H}}/\sqrt{n}$.

This yields

$$\left| \mathbb{E}_h[\phi_k^S(h)] - \frac{1}{n} \sum_{i=1}^n \phi_k^S(h_i) \right|$$

$$\leq |\mathbb{E}_h\left[\phi(h)|h \in \mathcal{B}[c_k]\right]| \sqrt{\frac{m \ln(4\sqrt{nm}) + \ln(2/\delta)}{2n}} + \frac{\varsigma_k R_{\mathcal{H}}}{\sqrt{n}} \left(\frac{1}{n}\sum_{i=1}^{n} \mathbb{1}\{h_i \in \mathcal{B}[c_k]\}\right).$$

$\square$

### D.3 Proof of Theorem 3

In the proof of Theorem 3, we write $f(x) := f(x, w, e, \theta)$ when the dependency on $(w, e, \theta)$ is clear from the context.

**Lemma 1.** *Fix $\theta \in \Theta$ and $e \in E$. Then for any $\delta > 0$, with probability at least $1 - \delta$ over an iid draw of $n$ examples $((x_i, y_i))_{i=1}^{n}$, the following holds for any $w \in \mathcal{W}$:*

$$\left|\mathbb{E}_{x,y}[J(f(x, w, e, \theta), y)] - \frac{1}{n}\sum_{i=1}^{n} J(f(x, w, e, \theta), y_i)\right| \leq C_J(w)\sqrt{\frac{4L^G \ln 2 + 2\ln(1/\delta)}{n}}.$$

*Proof of Lemma 1.* Let $\mathcal{I}_{k,y} = \{i \in [n] : (q_e \circ h_\theta)(x_i) = Q_k, y_i = y\}$. Using $\mathbb{E}_{x,y}[J(f(x), y)] = \sum_{k=1}^{L^G}\sum_{y' \in \mathcal{Y}} \mathbb{E}_{x,y}[J(f(x), y)|(q_e \circ h_\theta)(x) = Q_k, y = y'] \Pr((q_e \circ h_\theta)(x) = Q_k \wedge y = y')$, we first decompose the difference into two terms as

$$\mathbb{E}_{x,y}[J(f(x), y)] - \frac{1}{n}\sum_{i=1}^{n} J(f(x_i), y_i) \tag{9}$$

$$= \sum_{k=1}^{L^G}\sum_{y' \in \mathcal{Y}} \mathbb{E}_{x,y}[J(f(x), y)|(q_e \circ h_\theta)(x) = Q_k, y = y']\left(\Pr((q_e \circ h_\theta)(x) = Q_k \wedge y = y') - \frac{|\mathcal{I}_{k,y'}|}{n}\right)$$

$$+ \left(\sum_{k=1}^{L^G}\sum_{y' \in \mathcal{Y}} \mathbb{E}_{x,y}[J(f(x), y)|(q_e \circ h_\theta)(x) = Q_k, y = y']\frac{|\mathcal{I}_{k,y'}|}{n} - \frac{1}{n}\sum_{i=1}^{n} J(f(x_i), y_i)\right).$$

The second term in the right-hand side of (9) is further simplified by using

$$\frac{1}{n}\sum_{i=1}^{n} J(f(x), y) = \frac{1}{n}\sum_{k=1}^{L^G}\sum_{y' \in \mathcal{Y}}\sum_{i \in \mathcal{I}_{k,y'}} J(f(x_i), y_i),$$

and

$$\mathbb{E}_{x,y}[J(f(x), y)|(q_e \circ h_\theta)(x)$$
$$= Q_k, y = y']\frac{|\mathcal{I}_{k,y'}|}{n} = \frac{1}{n}\sum_{i \in \mathcal{I}_{k,y'}} \mathbb{E}_{x,y}[J(f(x), y)|(q_e \circ h_\theta)(x) = Q_k, y = y'],$$

as

$$\sum_{k=1}^{L^G}\sum_{y' \in \mathcal{Y}} \mathbb{E}_{x,y}[J(f(x), y)|(q_e \circ h_\theta)(x) = Q_k, y = y']\frac{|\mathcal{I}_{k,y'}|}{n} - \frac{1}{n}\sum_{i=1}^{n} J(f(x_i), y_i)$$

$$= \frac{1}{n}\sum_{k=1}^{L^G}\sum_{y' \in \mathcal{Y}}\sum_{i \in \mathcal{I}_{k,y'}} (\mathbb{E}_{x,y}[J(f(x), y)|(q_e \circ h_\theta)(x) = Q_k, y = y'] - J(f(x_i), y_i))$$

$$= \frac{1}{n}\sum_{k=1}^{L^G}\sum_{y' \in \mathcal{Y}}\sum_{i \in \mathcal{I}_{k,y'}} (J(\varphi_w(Q_k), y') - (J(\varphi_w(Q_k), y')) = 0$$

Substituting these into equation (9) yields

$$\mathbb{E}_{x,y}[J(f(x), y)] - \frac{1}{n}\sum_{i=1}^{n} J(f(x_i), y_i)$$

$$= \sum_{k=1}^{L^G} \sum_{y' \in \mathcal{Y}} J(\varphi_w(Q_k), y') \left( \Pr((q_e \circ h_\theta)(x) = Q_k \wedge y = y') - \frac{|\mathcal{I}_{k,y'}|}{n} \right)$$

$$= \sum_{k=1}^{2L^G} J(v_k) \left( \Pr(((q_e \circ h_\theta)(x), y) = v_k) - \frac{|\mathcal{I}_k|}{n} \right),$$

where the last line uses the fact that $\mathcal{Y} = \{y^{(1)}, y^{(2)}\}$ for some $(y^{(1)}, y^{(2)})$, along with the additional notation $\mathcal{I}_k = \{i \in [n] : ((q_e \circ h_\theta)(x_i), y_i) = v_k\}$. Here, $v_k$ is defined as $v_k = (\varphi_w(Q_k), y^{(1)})$ for all $k \in [L^G]$ and $v_k = (\varphi_w(e_{k-L^G}), y^{(2)})$ for all $k \in \{L^G + 1, \ldots, 2L^G\}$.

By using the bound of $|J(\varphi_w(\eta), y)| \le C_J(w)$,

$$\left| \mathbb{E}_{x,y}[J(f(x), y)] - \frac{1}{n} \sum_{i=1}^n J(f(x_i), y_i) \right|$$

$$= \left| \sum_{k=1}^{2L^G} J(v_k) \left( \Pr(((q_e \circ h_\theta)(x), y) = v_k) - \frac{|\mathcal{I}_k|}{n} \right) \right|$$

$$\le C_J(w) \sum_{k=1}^{2L^G} \left| \Pr(((q_e \circ h_\theta)(x), y) = v_k) - \frac{|\mathcal{I}_k|}{n} \right|.$$

Since $|\mathcal{I}_k| = \sum_{i=1}^n \mathbb{1}\{((q_e \circ h_\theta)(x_i), y_i) = v_k\}$ and $(\theta, e)$ is fixed, the vector $(|\mathcal{I}_1|, \ldots, |\mathcal{I}_{2L^G}|)$ follows a multinomial distribution with parameters $n$ and $p = (p_1, \ldots, p_{2L^G})$, where $p_k = \Pr(((q_e \circ h_\theta)(x), y) = v_k)$ for $k = 1, \ldots, 2L^G$. Thus, by using the Bretagnolle-Huber-Carol inequality (van der Vaart and Wellner, 1996, A6.6 Proposition), we have that with probability at least $1 - \delta$,

$$\left| \mathbb{E}_{x,y}[J(f(x), y)] - \frac{1}{n} \sum_{i=1}^n J(f(x_i), y_i) \right| \le C_J(w) \sqrt{\frac{4L^G \ln 2 + 2 \ln(1/\delta)}{n}}.$$

$\square$

*Proof of Theorem 3.* Let $\hat{\mathcal{C}}_{r,d} \in \arg\min_{\mathcal{C}} \{ |\mathcal{C}| : \mathcal{C} \subseteq \mathcal{M}, E \times \Theta \subseteq \cup_{c \in \mathcal{C}} \mathcal{B}_{(\mathcal{M},d)}[c, r] \}$. Note that if $\mathcal{N}_{(\mathcal{M},d)}(r, E \times \Theta) = \infty$, the bound in the statement of the theorem vacuously holds. Thus, we focus on the case of $\mathcal{N}_{(\mathcal{M},d)}(r, E \times \Theta) = |\hat{\mathcal{C}}_{r,d}| < \infty$. For any $(w, e, \theta) \in \mathcal{W} \times E \times \Theta$, the following holds: for any $(\hat{e}, \hat{\theta}) \in \hat{\mathcal{C}}_{r,d}$,

$$|\psi_w(e, \theta)| = \left| \psi_w(\hat{e}, \hat{\theta}) + \psi_w(e, \theta) - \psi_w(\hat{e}, \hat{\theta}) \right|$$

$$\le \left| \psi_w(\hat{e}, \hat{\theta}) \right| + \left| \psi_w(e, \theta) - \psi_w(\hat{e}, \hat{\theta}) \right|. \tag{10}$$

For the first term in the right-hand side of (10), by using Lemma 1 with $\delta = \delta'/\mathcal{N}_{(\mathcal{M},d)}(r, E \times \Theta)$ and taking union bounds, we have that for any $\delta' > 0$, with probability at least $1 - \delta'$, the following holds for all $(\hat{e}, \hat{\theta}) \in \hat{\mathcal{C}}_{r,d}$,

$$\left| \psi_w(\hat{e}, \hat{\theta}) \right| \le C_J(w) \sqrt{\frac{4L^G \ln 2 + 2 \ln(\mathcal{N}_{(\mathcal{M},d)}(r, E \times \Theta)/\delta')}{n}}. \tag{11}$$

By combining equations (10) and (11), we have that for any $\delta' > 0$, with probability at least $1 - \delta'$, the following holds for any $(w, e, \theta) \in \mathcal{W} \times E \times \Theta$ and any $(\hat{e}, \hat{\theta}) \in \hat{\mathcal{C}}_{r,d}$:

$$|\psi_w(e, \theta)| \le C_J(w) \sqrt{\frac{4L^G \ln 2 + 2 \ln(\mathcal{N}_{(\mathcal{M},d)}(r, E \times \Theta)/\delta')}{n}} + \left| \psi_w(e, \theta) - \psi_w(\hat{e}, \hat{\theta}) \right|.$$

This implies that for any $\delta' > 0$, with probability at least $1 - \delta'$, the following holds for any $(w, e, \theta) \in \mathcal{W} \times E \times \Theta$:

$$|\psi_w(e, \theta)| \le C_J(w) \sqrt{\frac{4L^G \ln 2 + 2 \ln(\mathcal{N}_{(\mathcal{M},d)}(r, E \times \Theta)/\delta')}{n}} + \min_{(\hat{e}, \hat{\theta}) \in \hat{\mathcal{C}}_{r,d}} \left| \psi_w(e, \theta) - \psi_w(\hat{e}, \hat{\theta}) \right|. \tag{12}$$

For the second term in the right-hand side of (12), we have that for any $(w, e, \theta) \in \mathcal{W} \times E \times \Theta$,

$$\min_{(\hat{e}, \hat{\theta}) \in \hat{\mathcal{C}}_{r,d}} \left| \psi_w(e, \theta) - \psi_w(\hat{e}, \hat{\theta}) \right| \leq \mathcal{L}_d(w) \min_{(\hat{e}, \hat{\theta}) \in \hat{\mathcal{C}}_{r,d}} d((e, \theta), (\hat{e}, \hat{\theta})) \leq \mathcal{L}_d(w) r.$$

Thus, by using $r = \mathcal{L}_d(w)^{1/\rho - 1} \sqrt{\frac{1}{n}}$, we have that for any $\delta' > 0$, with probability at least $1 - \delta'$, the following holds for any $(w, e, \theta) \in \mathcal{W} \times E \times \Theta$:

$$|\psi_w(e, \theta)| \leq C_J(w) \sqrt{\frac{4L^G \ln 2 + 2\ln(\mathcal{N}_{(\mathcal{M},d)}(r, E \times \Theta)/\delta')}{n}} + \sqrt{\frac{\mathcal{L}_d(w)^{2/\rho}}{n}}. \qquad (13)$$

Since this statement holds for any $\delta' > 0$, this implies the statement of this theorem. $\qquad \square$

### D.4   Proof of Theorem 4

In the proof of Theorem 3, we write $\tilde{f}(x) := \tilde{f}(x, w, \theta)$ when the dependency on $(w, \theta)$ is clear from the context.

**Lemma 2.** *Fix $\theta \in \Theta$. Let $(\mathcal{M}', d')$ be a matric space such that $\mathcal{H} \subseteq \mathcal{M}'$. Fix $r' > 0$ and $\bar{\mathcal{C}}_{r',d'} \in \arg\min_{\mathcal{C}} \{|\mathcal{C}| : \mathcal{C} \subseteq \mathcal{M}', \mathcal{H} \subseteq \cup_{c \in \mathcal{C}} \mathcal{B}_{(\mathcal{M}',d')}[c, r']\}$. Assume that for any $c \in \bar{\mathcal{C}}_{r',d'}$, we have $|(J(\varphi_w(h), y) - (J(\varphi_w(h'), y)| \leq \xi(w, r', \mathcal{M}', d)$ for any $h, h' \in \mathcal{B}_{(\mathcal{M}',d')}[c, r']$ and $y \in \mathcal{Y}$. Then for any $\delta > 0$, with probability at least $1 - \delta$ over an iid draw of $n$ examples $((x_i, y_i))_{i=1}^n$, the following holds for any $w \in \mathcal{W}$:*

$$\left| \mathbb{E}_{x,y}[J(\tilde{f}(x, w, \theta), y)] - \frac{1}{n} \sum_{i=1}^n J(\tilde{f}(x, w, \theta), y_i) \right|$$

$$\leq \tilde{C}_J(w) \sqrt{\frac{4\mathcal{N}_{(\mathcal{M}',d')}(r', \mathcal{H}) \ln 2 + 2\ln(1/\delta)}{n}} + \xi(w, r', \mathcal{M}', d).$$

*Proof of Lemma 2.* Note that if $\mathcal{N}_{(\mathcal{M}',d')}(r', \mathcal{H}) = \infty$, the bound in the statement of the theorem vacuously holds. Thus, we focus on the case of $\mathcal{N}_{(\mathcal{M}',d')}(r', \mathcal{H}) = |\bar{\mathcal{C}}_{r',d'}| < \infty$. Fix an arbitrary ordering and define $c_k \in \bar{\mathcal{C}}_{r',d'}$ to be the $k$-the element in the ordered version of $\bar{\mathcal{C}}_{r',d'}$ in that fixed ordering (i.e., $\cup_k \{c_k\} = \bar{\mathcal{C}}_{r',d'}$).

Let $\mathcal{I}_{k,y} = \{i \in [n] : h_\theta(x_i) \in \mathcal{B}_{(\mathcal{M}',d')}[c_k, r'], y_i = y\}$ for all $k \times y \in [|\bar{\mathcal{C}}_{r',d'}|] \times \mathcal{Y}$. Using $\mathbb{E}_{x,y}[J(\tilde{f}(x), y)] = \sum_{k=1}^{|\bar{\mathcal{C}}_{r',d'}|} \sum_{y' \in \mathcal{Y}} \mathbb{E}_{x,y}[J(\tilde{f}(x), y)|h_\theta(x) \in \mathcal{B}_{(\mathcal{M}',d')}[c_k, r'], y = y'] \Pr(h_\theta(x) \in \mathcal{B}_{(\mathcal{M}',d')}[c_k, r'] \wedge y = y')$, we first decompose the difference into two terms as

$$\left| \mathbb{E}_{x,y}[J(\tilde{f}(x), y)] - \frac{1}{n} \sum_{i=1}^n J(\tilde{f}(x_i), y_i) \right| \qquad (14)$$

$$= \left| \sum_{k=1}^{|\bar{\mathcal{C}}_{r',d'}|} \sum_{y' \in \mathcal{Y}} \mathbb{E}_{x,y}[J(\tilde{f}(x), y)|h_\theta(x) \in \mathcal{B}_{(\mathcal{M}',d')}[c_k, r'], y = y'] \left( \Pr(h_\theta(x) \in \mathcal{B}_{(\mathcal{M}',d')}[c_k, r'] \wedge y = y') - \frac{|\mathcal{I}_{k,y'}|}{n} \right) \right|$$

$$+ \left| \sum_{k=1}^{|\bar{\mathcal{C}}_{r',d'}|} \sum_{y' \in \mathcal{Y}} \mathbb{E}_{x,y}[J(\tilde{f}(x), y)|h_\theta(x) \in \mathcal{B}_{(\mathcal{M}',d')}[c_k, r'], y = y'] \frac{|\mathcal{I}_{k,y'}|}{n} - \frac{1}{n} \sum_{i=1}^n J(\tilde{f}(x_i), y_i) \right|.$$

The second term in the right-hand side of (14) is further simplified by using

$$\frac{1}{n} \sum_{i=1}^n J(\tilde{f}(x), y) = \frac{1}{n} \sum_{k=1}^{|\bar{\mathcal{C}}_{r',d'}|} \sum_{y' \in \mathcal{Y}} \sum_{i \in \mathcal{I}_{k,y'}} J(\tilde{f}(x_i), y_i),$$

and

$$\mathbb{E}_{x,y}[J(\tilde{f}(x), y)|h_\theta(x) \in \mathcal{B}_{(\mathcal{M}',d')}[c_k, r'], y = y'] \frac{|\mathcal{I}_{k,y'}|}{n}$$

$$= \frac{1}{n} \sum_{i \in \mathcal{I}_{k,y'}} \mathbb{E}_{x,y}[J(\tilde{f}(x), y)|h_\theta(x) \in \mathcal{B}_{(\mathcal{M}',d')}[c_k, r'], y = y'],$$

as

$$\left| \sum_{k=1}^{|\bar{\mathcal{C}}_{r',d'}|} \sum_{y' \in \mathcal{Y}} \mathbb{E}_{x,y}[J(\tilde{f}(x), y) | h_\theta(x) \in \mathcal{B}_{(\mathcal{M}',d')}[c_k, r'], y = y'] \frac{|\mathcal{I}_{k,y'}|}{n} - \frac{1}{n} \sum_{i=1}^{n} J(\tilde{f}(x_i), y_i) \right|$$

$$= \left| \frac{1}{n} \sum_{k=1}^{|\bar{\mathcal{C}}_{r',d'}|} \sum_{y' \in \mathcal{Y}} \sum_{i \in \mathcal{I}_{k,y'}} \left( \mathbb{E}_{x,y}[J(\tilde{f}(x), y) | h_\theta(x) \in \mathcal{B}_{(\mathcal{M}',d')}[c_k, r'], y = y'] - J(\tilde{f}(x_i), y_i) \right) \right|$$

$$\leq \frac{1}{n} \sum_{k=1}^{|\bar{\mathcal{C}}_{r',d'}|} \sum_{y' \in \mathcal{Y}} \sum_{i \in \mathcal{I}_{k,y'}} \sup_{h \in \mathcal{B}_{(\mathcal{M}',d')}[c_k,r']} |J(\varphi_w(h, y'), J(\varphi_w(h_\theta(x_i)), y')| \leq \xi(w).$$

Substituting these into equation (14) yields

$$\left| \mathbb{E}_{x,y}[J(\tilde{f}(x), y)] - \frac{1}{n} \sum_{i=1}^{n} J(\tilde{f}(x_i), y_i) \right|$$

$$\leq \left| \sum_{k=1}^{|\bar{\mathcal{C}}_{r',d'}|} \sum_{y' \in \mathcal{Y}} \mathbb{E}_{x,y}[J(\tilde{f}(x), y') | h_\theta(x) \in \mathcal{B}_{(\mathcal{M}',d')}[c_k, r']] \left( \Pr(h_\theta(x) \in \mathcal{B}_{(\mathcal{M}',d')}[c_k, r'] \wedge y = y') - \frac{|\mathcal{I}_{k,y'}|}{n} \right) \right| + \xi(w)$$

$$\leq \tilde{C}_J(w) \sum_{k=1}^{2|\bar{\mathcal{C}}_{r',d'}|} \left| \left( \Pr((h_\theta(x), y) \in v_k) - \frac{|\mathcal{I}_k|}{n} \right) \right| + \xi(w),$$

where the last line uses the fact that $\mathcal{Y} = \{y^{(1)}, y^{(2)}\}$ for some $(y^{(1)}, y^{(2)})$, along with the additional notation $\mathcal{I}_k = \{i \in [n] : (h_\theta(x_i), y_i) \in v_k\}$. Here, $v_k$ is defined as $v_k = \mathcal{B}_{(\mathcal{M}',d')}[c_k, r'] \times \{y^{(1)}\}$ for all $k \in [|\bar{\mathcal{C}}_{r',d'}|]$ and $v_k = \mathcal{B}_{(\mathcal{M}',d')}[c_{k-|\bar{\mathcal{C}}_{r',d'}|}, r'] \times \{y^{(2)}\}$ for all $k \in \{|\bar{\mathcal{C}}_{r',d'}| + 1, \ldots, 2|\bar{\mathcal{C}}_{r',d'}|\}$.

Since $|\mathcal{I}_k| = \sum_{i=1}^{n} \mathbb{1}\{(h_\theta(x), y) \in v_k\}$ and $\theta$ is fixed, the vector $(|\mathcal{I}_1|, \ldots, |\mathcal{I}_{2|\bar{\mathcal{C}}_{r',d'}|}|)$ follows a multinomial distribution with parameters $n$ and $p = (p_1, ..., p_{2|\bar{\mathcal{C}}_{r',d'}|})$, where $p_k = \Pr((h_\theta(x), y) \in v_k)$ for $k = 1, \ldots, 2|\bar{\mathcal{C}}_{r',d'}|$. Thus, by noticing $|\bar{\mathcal{C}}_{r',d'}| = \mathcal{N}_{(\mathcal{M}',d')}(r', \mathcal{H})$ and by using the Bretagnolle-Huber-Carol inequality (van der Vaart and Wellner, 1996, A6.6 Proposition), we have that with probability at least $1 - \delta$,

$$\left| \mathbb{E}_{x,y}[J(\tilde{f}(x), y)] - \frac{1}{n} \sum_{i=1}^{n} J(\tilde{f}(x_i), y_i) \right|$$

$$\leq \tilde{C}_J(w) \sqrt{\frac{4\mathcal{N}_{(\mathcal{M}',d')}(r', \mathcal{H}) \ln 2 + 2 \ln(1/\delta)}{n}} + \xi(w).$$

$\square$

*Proof of Theorem 4.* Let $\hat{\mathcal{C}}_{r,d} \in \arg\min_{\mathcal{C}} \{|\mathcal{C}| : \mathcal{C} \subseteq \mathcal{M}, \Theta \subseteq \cup_{c \in \mathcal{C}} \mathcal{B}_{(\mathcal{M},d)}[c, r]\}$. Note that if $\mathcal{N}_{(\mathcal{M},d)}(r, \Theta) = \infty$, the bound in the statement of the theorem vacuously holds. Thus, we focus on the case of $\mathcal{N}_{(\mathcal{M},d)}(r, \Theta) = |\hat{\mathcal{C}}_{r,d}| < \infty$. For any $(w, \theta) \in \mathcal{W} \times \Theta$, the following holds: for any $\hat{\theta} \in \hat{\mathcal{C}}_{r,d}$,

$$\left| \tilde{\psi}_w(\theta) \right| = \left| \tilde{\psi}_w(\hat{\theta}) + \tilde{\psi}_w(\theta) - \tilde{\psi}_w(\hat{\theta}) \right|$$

$$\leq \left| \tilde{\psi}_w(\hat{\theta}) \right| + \left| \tilde{\psi}_w(\theta) - \tilde{\psi}_w(\hat{\theta}) \right|. \tag{15}$$

For the first term in the right-hand side of (15), by using Lemma 2 with $\delta = \delta'/\mathcal{N}_{(\mathcal{M}',d')}(r', \Theta)$ and taking union bounds, we have that for any $\delta' > 0$, with probability at least $1 - \delta'$, the following holds for all $\hat{\theta} \in \hat{\mathcal{C}}_{r,d}$,

$$\left| \tilde{\psi}_w(\hat{\theta}) \right| \leq \tilde{C}_J(w) \sqrt{\frac{4\mathcal{N}_{(\mathcal{M}',d')}(r', \mathcal{H}) \ln 2 + 2 \ln(\mathcal{N}_{(\mathcal{M},d)}(r, \Theta)/\delta')}{n}} + \xi(w, r', \mathcal{M}', d). \tag{16}$$

By combining equations (15) and (16), we have that for any $\delta' > 0$, with probability at least $1 - \delta'$, the following holds for any $(w, \theta) \in \mathcal{W} \times \Theta$ and any $\hat{\theta} \in \hat{\mathcal{C}}_{r,d}$:

$$\left| \tilde{\psi}_w(\theta) \right|$$

$$\leq \tilde{C}_J(w) \sqrt{\frac{4\mathcal{N}_{(\mathcal{M}',d')}(r', \mathcal{H}) \ln 2 + 2 \ln(\mathcal{N}_{(\mathcal{M},d)}(r, \Theta)/\delta')}{n}} + \left| \tilde{\psi}_w(\theta) - \tilde{\psi}_w(\hat{\theta}) \right| + \xi(w, r', \mathcal{M}', d).$$

This implies that for any $\delta' > 0$, with probability at least $1 - \delta'$, the following holds for any $(w, \theta) \in \mathcal{W} \times \Theta$:

$$\left| \tilde{\psi}_w(\theta) \right| \leq C_J(w) \sqrt{\frac{4\mathcal{N}_{(\mathcal{M}',d')}(r', \mathcal{H}) \ln 2 + 2 \ln(\mathcal{N}_{(\mathcal{M},d)}(r, \Theta)/\delta')}{n}} \tag{17}$$

$$+ \min_{\hat{\theta} \in \hat{\mathcal{C}}_{r,d}} \left| \tilde{\psi}_w(\theta) - \tilde{\psi}_w(\hat{\theta}) \right| + \xi(w, r', \mathcal{M}', d).$$

For the second term in the right-hand side of (17), we have that for any $(w, \theta) \in \mathcal{W} \times \Theta$,

$$\min_{\hat{\theta} \in \hat{\mathcal{C}}_{r,d}} \left| \tilde{\psi}_w(\theta) - \tilde{\psi}_w(\hat{\theta}) \right| \leq \tilde{\mathcal{L}}_d(w) \min_{\hat{\theta} \in \hat{\mathcal{C}}_{r,d}} d(\theta, \hat{\theta}) \leq \mathcal{L}_d(w) r.$$

Thus, by using $r = \tilde{\mathcal{L}}_d(w)^{1/\rho - 1} \sqrt{\frac{1}{n}}$, we have that for any $\delta' > 0$, with probability at least $1 - \delta'$, the following holds for any $(w, \theta) \in \mathcal{W} \times \Theta$:

$$\left| \tilde{\psi}_w(\theta) \right| \leq \tilde{C}_J(w) \sqrt{\frac{4\mathcal{N}_{(\mathcal{M}',d')}(r', \mathcal{H}) \ln 2 + 2 \ln(\mathcal{N}_{(\mathcal{M},d)}(r, \Theta)/\delta')}{n}} \tag{18}$$

$$+ \sqrt{\frac{\tilde{\mathcal{L}}_d(w)^{2/\rho}}{n}} + \xi(w, r', \mathcal{M}', d).$$

Since this statement holds for any $\delta' > 0$, this implies the statement of this theorem. $\qquad \square$

## E  Method Details

---

**Algorithm 1:** Discretization of inter-module communication in RIM

---

N is sample size,T is total time step, M is number of modules in the RIM model

initialization;

**for** *i in 1..M* **do**
  $\quad$ initialize $z_i^0$;
**end**

Training;

**for** *n in 1..N* **do**
  $\quad$ **for** *t in 1..T* **do**
  $\quad\quad$ INPUTATTENTION $=$ SOFTATTENTION$(z_1^t, z_2^t, ..., z_M^t, x^t)$;
  $\quad\quad$ **if** *i in top K of* INPUTATTENTION **then**
  $\quad\quad\quad$ $\hat{z}_i^{t+1} = \text{RNN}(z_i^t, x^t)$;
  $\quad\quad$ **else**
  $\quad\quad\quad$ $\hat{z}_{i'}^{t+1} = z_{i'}^t$;
  $\quad\quad$ **end**
  $\quad\quad$ **for** *i in 1..M* **do**
  $\quad\quad\quad$ Discretization; $h_i^{t+1} = $ SOFTATTENTION$(\hat{z}_1^{t+1}, \hat{z}_2^{t+1}, ....\hat{z}_M^{t+1})$
  $\quad\quad\quad$ $z_i^{t+1} = \hat{z}_i^{t+1} + q(h_i^{t+1}, L, G)$;
  $\quad\quad$ **end**
  $\quad$ **end**
  $\quad$ Calculate task loss, codebook loss and commitment loss according to equation 1
  $\quad$ Update model parameter $\Theta$ together with discrete latent vectors in codebook $e \in R^{LXD}$;
**end**

---

### E.1 Task Details

2D shape environment is a 5X5 grid world with different objects of different shapes and colors placed at random positions.Each location can only be occupied by one object.The underlying environment dynamics of 3D shapes are the same as in the 2D dateset, and only the rendering component was changed (Kipf et al., 2019). In OOD setting, the total number of objects are changed for each environment. We used number of objects of 4 (validation), 3 (OOD-1) and 2 (OOD-2). We did not put in more than 5 objects because the environment will be too packed and the objects can hardly move.

The 3-body physics simulation environment is an interacting system that evolves according to physical laws.There are no actions applied onto any objects and movement of objects only depend on interaction among objects. This environment is adapted from Kipf et al. (2019). In the training environment, the radius of each ball is 3.In OOD settings, we changed the radius to 4 ( validation) and 2 (OOD test).

In all the 8 Atari games belong to the same collections of 2600 games from Atari Corporation.We used the games adapted to OpenAI gym environment. There are several versions of the same game available in OpenAI gym. We used version "Deterministic-v0" starting at warm start frame 50 for each game for training. Version "Frameskip-v0" starting at frame 250 as OOD validation and "Frameskip-v4" starting at frame 150 at OOD test.

In all the GNN compositional reasoning experiments. HITS at RANK K (K=1 in this study) was used as as the metrics for performance.This binary score is 1 for a particular example if the predicted state representation is in the k-nearest neighbor set around the true observation. Otherwise this score is 0. MEAN RECIPROCAL RANK (MRR) is also used as a performance metrics, which is defined as $MRR = \frac{1}{N} \sum_{n=1}^{N} \frac{1}{Rank_n}$ where $rank_n$ is the rank of the n-th sample (Kipf et al. (2019)).

In adding task, gap length of 500 was used for training and gap length of 200 (OOD validation) and 1000 (OOD testing) are used for OOD settings. In sequential MNIST experiment , model was trained at 14X14 resolution and tested in different resolutions (Goyal et al. (2019)). Sort-of-Clevr experiments are conducted in the same way as Goyal et al. (2021b)

### E.2 Model Architecture, Hyperparameters and Training Details

**DVNC implementation details**  In DVNC, codebook $e \in \mathbb{R}^{L \times m}$ was initialized by applying K-means clustering on training data points ($s$) where the number of clusters is $L$. The nearest $e_j$,by Euclidean distance ,is assigned to each $s_i$. The commitment loss $\beta \sum_i^G ||s_i - \text{sg}(e_{o_i})||_2^2$ , which encourages $s_i$ stay close to the chosen codebook vector, and the task loss are back-propagated to each of the model components that send information in the inter-component communication process.The gradient of task loss are back-propagated to each of the components that send information using straight-through gradient estimator. The codebook loss $\sum_i^G ||\text{sg}(s_i) - e_{o_i}||^2$ that encourages the selected codebook vector to stay close to $s_i$ is back-propagated to the selected codebook vector. Task loss is not backpropagated to codebook vectors. Only the task loss is back-propagated to the model component that receives the information. It is worth pointing out that in this study, we train the codebook vectors directly using gradient descent instead of using exponential moving average updates as in Oord et al. (2017).

Model architecture, hyperparameters and training settings of GNN used in this study are same as in Kipf et al. (2019), where encoder dimension is 4 and number of object slot is 3..Model architecture, hyperparameters and training settings of RIMs used in this study are identify to Goyal et al. (2019),where 6 RIM units and k=4 are used. Model architecture, hyperparameters and training settings of transformer models are the same as in Goyal et al. (2021b), except that we did not include shared workspace. Hyperparameters of GNN and RIM models are summarized in table 4. Hyperparameters of transformers with various settings can be found in Goyal et al. (2021b). In all the models mentioned above,we include discretization of communication in DVNC and keep other parts of the model unchanged.

Data are split into training set, validation set and test set, the ratio varies among different tasks depending on data availability.For in-distribution performance, validation set has the same distribution as training set. In OOD task, one of the OOD setting,eg. certain number of blocks in 2D shape

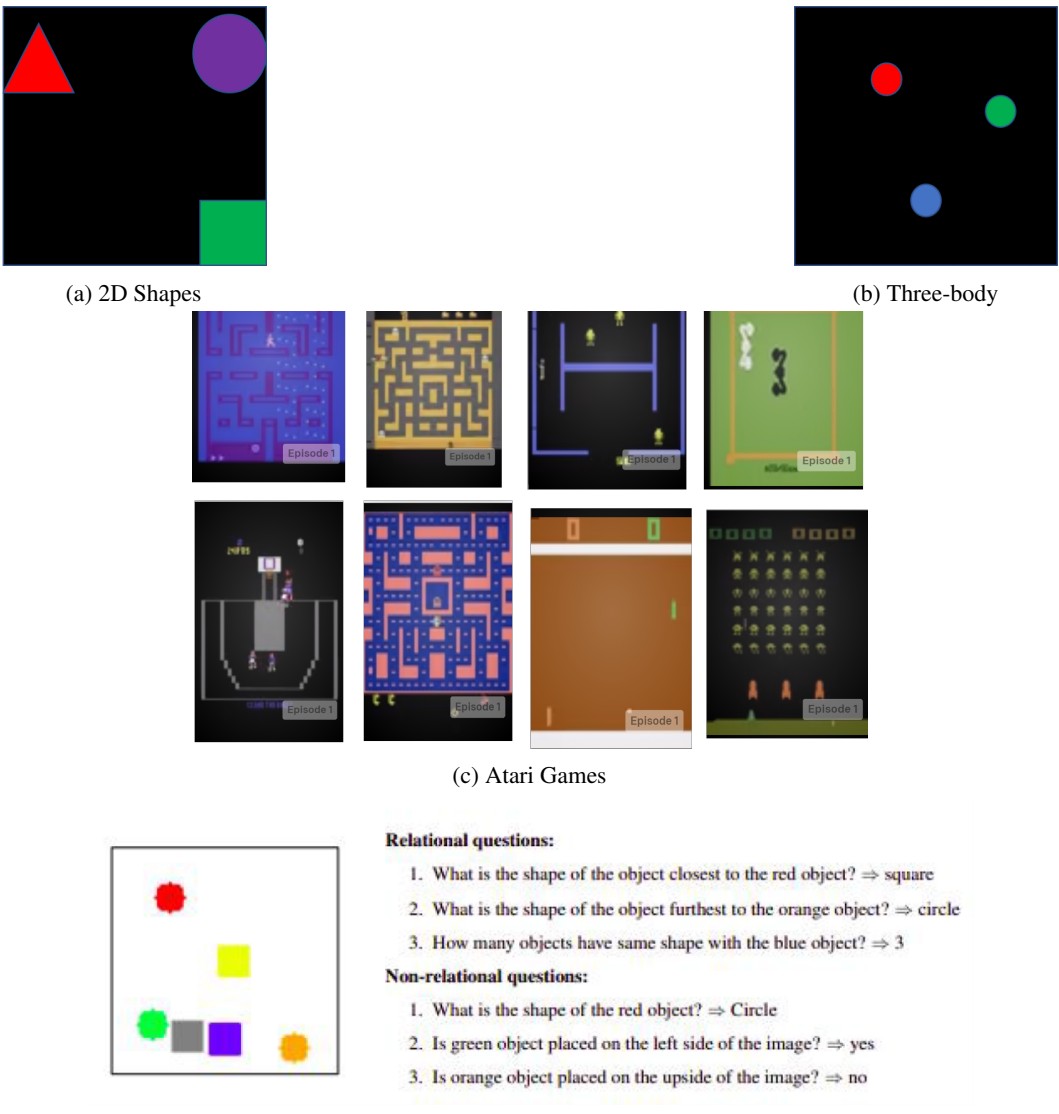

(a) 2D Shapes         (b) Three-body

(c) Atari Games

**Relational questions:**

1. What is the shape of the object closest to the red object? ⇒ square
2. What is the shape of the object furthest to the orange object? ⇒ circle
3. How many objects have same shape with the blue object? ⇒ 3

**Non-relational questions:**

1. What is the shape of the red object? ⇒ Circle
2. Is green object placed on the left side of the image? ⇒ yes
3. Is orange object placed on the upside of the image? ⇒ no

(d) Sort-of-Clevr

Figure 7: Examples of different task environments. Atari game screen shots are obtained from OpenAI gym platform. Sort-of-Clevr example was adapted from Goyal et al. (2021b) with permission

Table 4: Hyperparameters used for GNN and RIMs

| GNN model | | | RIMs model | |
|---|---|---|---|---|
| | | | | |
| Hyperparameters | Values | | Hyperparameters | Values |
| Batch size | 1024 | | Batch size | 64 |
| hidden dim | 512 | | hidden dim | 300 |
| embedding-dim | 512/G | | embedding-dim | 300/G |
| codebook_loss_weight | 1 | | codebook_loss_weight | 0.25 |
| Max. number of epochs | 200 | | Max. number of epochs | 100 |
| Number of slots(objects) | 5 | | learning-rate | 0.001 |
| learning-rate | 5.00E-04 | | Optimizer | Adam |
| Optimizer | Adam | | Number of Units (RIMs) | 6 |
| | | | Number of active RIMs | 4 |
| | | | RIM unit type | LSTM |
| | | | dropout | 0.5 |
| | | | gradient clipping | 1 |

experiment, is used as validation set. The OOD setting used for validation was not included in test set.

## E.3  Computational resources

GPU nodes on university cluster are used. GNN training takes 3 hrs for each task with each hyperparameter setting on Tesla GPU. Training of RIMs and transformers take about 12 hours on the same GPU for each task. In total, the whole training progress of all models, all tasks, all hyperparameter settings takes approximately 800 hours on GPU nodes.