# OpenReview forum: "Discrete-Valued Neural Communication"
_NeurIPS.cc/2021/Conference — NeurIPS 2021 Poster_

### Official Review · Reviewer_X9GZ · 2021-06-28

**Rating:** 7
**Confidence:** 4

**Summary:**

This paper proposes a method of discretizing communication between different modules in a network of specialists in order to improve generalization. They propose using a common codebook of discrete symbols shared by all network components that can be used to share information between modules. They propose a method of discretization based on the VQVAE architecture called Discrete-Valued Neural Communication (DVNC), in which a continuous vector is partitioned into "discretization heads", each of which is then mapped to its nearest neighbour among a set of learned latent vectors that are shared between modules. Gradients are propagated through the discretization using a straight-through estimator.
The authors provide theoretical analysis to argue that DVNC improves sample efficiency and robustness to noise, while decreasing intrinsic dimensionality. They also provide experiments with a number of different tasks and architectures to show that their method of discretization improves performance on out-of-distribution samples at inference time.

**Limitations And Societal Impact:**

As this is a purely technical and theoretical work, the direct negative social impacts are not a major concern. The authors do address the limitations of their work, though I think it would be beneficial to more thoroughly address the concerns that discretization could harm expressivity or reduce in-distribution performance (perhaps with more robust experiments to show that this is empirically not the case).

**Main Review:**

This paper’s analysis of the benefits of a discrete latent space for model generalization is novel and valuable, and highlights an interesting avenue for future research. The mathematics and theoretical results presented are thorough and rigorous, and provide a very interesting look at the benefits of discretization for generalization in general terms. The authors provide evidence for their claims through a variety of experimental settings, backed up by theoretical analysis and examples using Gaussian-distributed vectors. The work is clearly novel and provides a new and relevant framework for analysis that will hopefully be built upon by future research.

My main criticism of the paper would be that I think it needs some additional work in editing and improving the clarity of some sections, particularly the theoretical analysis in Section 2. From reading the paper it is not entirely clear to me how the results stated in Theorems 1 & 2  generalize to the claims about noise robustness and intrinsic dimensionality made in the text. First of all, in looking at the theorems, my impression is that they are discussing only functions restricted to act on a small specific subdomain of the input space, and I don't see from the content of the text or the theorems any justification for the more general claims you make. From your response I gather this does generalize, but I would make the generalization more clear. Second, I am unclear on what the paper means by 'noise robustness'. As far as I can tell, there is no explicit noise model mentioned in the paper or the appendices. The theorems rather appear to be analyzing the error of approximating the output of functions by Monte Carlo sampling. Finally, the paper’s discussion of ‘intrinsic dimensionality’ could benefit from some additional detail and discussion, as the definition or significance of this was never made clear in the paper (is this a reference to the Minkowski dimension, or maybe the packing dimension proposed by Kegl?). It would be helpful to have a more concrete discussion of what the intended meaning of these terms is in the context of the two theorems.

I also think that the experiments may be overly limited. While the paper shows experiments on a number of tasks and architectures, many seem to be toy tasks rather than more realistic applications. In addition, aside from Figure 4, the reported results (e.g. Table 2, Table 3, Figure 5) show only the effect of the method on the out of distribution performance, and do not show the effect of the method on in-distribution performance, which I think would be beneficial to include for completeness’s sake.

There are also some clear editing mistakes in the paper that need to be fixed (e.g. ‘selected’ repeated on page 3, appendix B missing, etc…). Many of the diagrams are also far too small, making them very difficult to read – particularly figure 4.

Overall I think that this paper is a worthwhile and relevant work, as long as the issues with clarity and editing are fixed.


**Time Spent Reviewing:**

5

---

> ### Author Response · Authors · 2021-08-10
> **Follow-Ups on Theory**
>
> We appreciate the positive feedback from the reviewer. We will adjust the manuscript’s theorem section to make it easier for readers to link the conclusion of theoretical analysis to other parts of the manuscript and different claims. We will extend the explanation of “noise robustness” and several other terms in the theoretical section. The figures and table will be adjusted to make them more readable. The editing errors will be corrected and image font sizes will also be increased in the manuscript.
>
> We will add more text in section 2 to explain that results from Theorems 1 & 2 implies the conclusion that discretization leads to increased noise robustness and reduced intrinsic dimensionality. Theorem 1 and 2 apply to all internal representations in a model, including but not limited to results of attention mechanisms and edges of GNN.
>
> *“my impression is that they are discussing only functions restricted to act on a small specific subdomain of the input space.”* Please note that this restricted claim trivially applies to the whole domain, following from the sum over subdomains, $\phi(h) = \sum_k \phi_k^S (h)$. Moreover, we have the corresponding theorems for the entire domain in Theorems 3 and 4 In Appendix A with further discussions and explanations in Appendix C. Therefore, our claims hold and follow from the results in our paper. However, we admit that this could be unclear and we added the above explanations in the revised manuscript.
>
> As for the term noise robustness, we realize the term (see equations 2 and 3) causes confusion and have re-named it *sensitivity bounds*.  We define the (noise) sensitivity term to be the second term on the right hand side of equation in Theorem 2 (and the third term on the right hand side of equation in Theorem 4). This term measures how the model can change as we inject small noises. We revised the paper accordingly to clarify this point. Thus, in our theory, it is shown that one of the benefits of the discretization is that it can make the (noise) sensitivity term to be zero in the error bounds.
>
> As for the intrinsic dimensionality, it is defined in the last paragraph of the first section (the introduction section) as “the logarithm of the covering number””[ref. 1]. . When we explained the logarithm of the covering number to people without math background, we faced the difficulty to communicate the concept without taking up space to fit in the page limit. We also obtained suggestions to name it with some intuitive name, which we followed. We are open to any suggestions here. For now, we replaced the name of the intrinsic dimensionality with the term metric entropy, which is indeed the logarithm of the covering number and thus would be a better name here.
>
> As for the intrinsic dimensionality (or the metric entropy in the revised version), in our theory, it is shown that another benefit of the discretization (in addition to the reduction on the noise sensitivity term) is that it can reduce the the intrinsic dimensionality (or the metric entropy), as shown in Theorem 1 (and Theorem 3).
>
> As suggested by the reviewer , we will include all in-distribution results. For example, the in-distribution performance of DVNC vs. baseline is 0.2912 vs 0.2935 in “Alien”, 0.8552 vs. 0.8678 in “BankHeis”, 0.6433 vs. 0.6510 in “Berzerk”, 0.9501 vs. 0.9501 in “Boxing”, 0.8891 vs. 0.9001 in “DoubleDunk” , 0.2411 vs. 0.2918 in “MsPacman”, 0.3034 vs. 0.2998 in  “Pong”, 0.2101 vs. 0.2401 in “SpaceInvaders”
>
>  As for figure 5, the purpose of the figure is to show effects of different values of G and L on OOD generalization, which corresponds to our theoretical analysis, and compare OOD generalization when different parts of the model are discretized. Therefore, we think it is a bit misleading if also including in-distribution results in the figures as they will have different meanings.
>
>
> [1]http://www.stat.yale.edu/~yw562/teaching/598/lec14.pdf

---

> > ### Comment · Reviewer_X9GZ · 2021-08-11
> > **This largely addresses my concerns.**
> >
> > Thank you for the response.
> >
> > I assumed the results from the theorems would generalize to a larger class of input functions in some way similar to what you suggest, but I think making that generalization more explicit would be helpful for readability. Similarly, while I understand that it is difficult to communicate the concepts while staying within the page limit, the 'intrinsic dimensionality' you refer to is mentioned frequently enough that I think it does need to be specifically discussed. Vis a vis the noise sensitivity, I think it would be helpful to be more clear about what sort of noise or small pertubations you are discussing, as I did not find that obvious from the reading.
> >
> > As long as those things are done and the other changes you mentioned are made, I think my concerns are addressed.

---

> > > ### Author Response · Authors · 2021-08-12
> > > **More Notes on Theory**
> > >
> > > For the intrinsic dimensionality, we revised the paper by changing its name to the "metric entropy", which is the logarithm of the covering number. For the concepts of the covering number and the metric entropy, we added a new reference (http://www.stat.yale.edu/~yw562/teaching/598/lec14.pdf). To improve connections to other parts of the manuscript, we added an explanation of how the metric entropy and covering number can be used to bound the generalization error of learning functions.  Also, how the metric entropy plays the role of VC dimension in classification. For these points, we added references [1-3]. However, please note that our use of the metric entropy is very different from how it is used in these references and the standard statistical learning theory. The main difference is the following: the metric entropy in the standard learning theory [1-3] is used for the model space or hypothesis space. This leads to a large bound for a large capacity model space. In contrast, our analysis avoids the dependence on the model capacity, by using the metric entropy on the bottleneck, instead of the model space. As a result, our bound is totally free from the model capacity after the bottleneck: e.g., even for cases where the Rademacher complexity (or VC dimension) of the model space (after the bottleneck) is unboundedly large or vacuously large, we provide good finite bounds in Theorem 1 (and Theorem 3). We believe that this is an interesting contribution.
> > >
> > > For the generalizability, note that the function φ is an arbitrary map, including any function mapping h to an evaluation criterion in these examples. We can add this sentence to the revised paper.
> > >
> > > Please note that for our theory, we do not make any assumption on the specific type of noise injected (e.g. gaussian noise). Instead, the noise sensitivity term naturally arises as a consequence of the learning theory analysis. More concretely, the (noise) sensitivity term measures how the model can change w.r.t. any noise that maximizes the change in the bounded domain of radius propositional to R_H.
> > >
> > > - [1] https://www.jmlr.org/papers/volume2/zhang02b/zhang02b.pdf
> > > - [2] Mohri, Mehryar; Rostamizadeh, Afshin; Talwalkar, Ameet (2012). Foundations of Machine Learning. USA, Massachusetts: MIT Press. ISBN 9780262018258.
> > > - [3] https://ocw.mit.edu/courses/mathematics/18-657-mathematics-of-machine-learning-fall-2015/lecture-notes/MIT18_657F15_L6.pdf

---

### Official Review · Reviewer_ypgC · 2021-07-13

**Rating:** 2
**Confidence:** 4

**Summary:**

The submission discusses discrete communication between neural network components and proposes 'discrete-valued neural communication (DVNC)', i.e. discretization of continuous representations in neural networks. The key contribution is a segmented discretization of vectors (see Eq. 1), in contrast to a joint discretization in a prior work (Oord et al., 2017).

**Main Review:**

The submission addresses an important research questions, namely how different neural submodules can communicate effectively. This is relevant since future neural networks may have several specialized components that need to interact in an efficient and effective way. However, the submission does not discuss clearly related prior work. Furthermore, is remains unclear what the paper actually claims to contribute. Hence, I believe that the submission cannot be accepted to this conference in its current form.

More details:

Missing/vague contribution:
One of the main reasons for my reject recommendation is the vagueness of the submission. The title 'Discrete-Valued Neural Communication' and the abstract/introduction suggest that this paper proposes discrete-valued neural communication for the first time. However, this is clearly not the case. There are several works that have been working on this area (see details below). Hence, after reading the first parts of the submission, it is unclear what its actual contribution is. The actual contribution becomes more clear in section 2, and specifically in Equation 1, which is very similar to Equation 3 in the VQ-VAE paper (Oord et al., 2017). The key difference is the segmentation and the separate discretization of h. However, this does not become clear at all in the introduction of the submission. I would like to recommend to clearly state early on what the actual contribution of the paper is instead of vaguely talking about discrete communication.

Missing related work:
Discrete-valued neural communication has already been addressed by several lines of work. Unfortunately, non of these areas is discussed. Examples are discrete communication between different agents in reinforcement learning (e.g. [1]), networks with discrete-continuous computation graphs (e.g. [2]), and memory networks with discrete states (e.g. [3]). Hence, I would like to recommend to discuss these works and how the current submission differs.

[1] Jakob N. Foerster et al. Learning to Communicate with Deep Multi-Agent Reinforcement Learning. In: CoRR abs/1605.06676 (2016). arXiv: 1605.06676. url: http://arxiv.org/abs/1605.06676.
[2] Schulman, J., Heess, N., Weber, T., & Abbeel, P. (2015). Gradient Estimation Using Stochastic Computation Graphs. Advances in Neural Information Processing Systems, 3528–3536.
[3] Wang, C., & Niepert, M. (2019). State-Regularized Recurrent Neural Networks. Proceedings of the 36th International Conference on Machine Learning, 6596–6606. http://proceedings.mlr.press/v97/wang19j.html

Minor important comments/suggestions:
- The submission states that 'graph neural networks are composed of distinct nodes'. I think this description is rather confusing. A GNN is not composed of distinct nodes. It operates over nodes as underlying structure. Similarly, an RNN for NLP is not composed of tokens, but operates on a sequence of tokens. Similarly, transformers are not composed of positional elements.

- The submission argues that discrete encoding have an evolutionary advantage since different areas of the human brain are tuned to discrete variables. However, making strong connections between human brains and neural networks is far fetched. It remains an hypothesis if computers may benefit from similarities with the human brain.

- There are several typos and missing spaces in the 'Communication among Specialists' section.

- The text in Figure 4 should be larger. It is impossible to read it without zooming in.

**Time Spent Reviewing:**

3

---

> ### Author Response · Authors · 2021-08-10
> **Connections to Related Work and Contributions**
>
> We appreciate the insightful and valuable feedback in the review.  We apologize for the insufficiently precise description of the novel contributions in the early part of the manuscript. The reviewer finds our contribution to be "vague" because we use the introduction of the paper to situate our work in the historical context of machine learning and to focus on a fundamental issue in the field (symbols versus statistics). We view it as a positive that we adopt this broad perspective, but appreciate that the reviewer and other readers have come to expect an "introduction" to contain a summary of results. We will accordingly edit the introduction to include the contribution of this work to the community, but ask the reviewer to consider the possibility that other readers will appreciate the broad framing of the work.
>
> We would also emphasize that the novelty of our work is the use of discretization (with a mechanism similar to VQ-VAE) to regularize the information communicated between specialist subsystems, for example modules which communicate using attention.  While we could have done a better job of discussing related work, and we will update our manuscript to improve this discussion, we believe that the other papers you mentioned are fairly different in focus from that of our work, since they do not consider an architecture for discretizing the results of attention (and communication more generally).
>
> *“Discrete-valued neural communication has already been addressed by several lines of work… State-Regularized RNN and Learning to Communicate in Deep RL.”*
>
> We use the term communication to refer to the information sharing between distinct specialists.  The State-Regularized RNN paper is different because it considers discretization of the hidden states of an RNN, and not the communication mechanism.  The “Learning to Communicate in Deep RL” paper is related but focused specifically on the RL domain and did not consider more general use within DL architectures.  Moreover, they either learned communication using credit-assignment based on RL or they sent real-valued signals during training: “during learning the agents are free to send real-valued messages to each other”.  Thus while we should cite their technique, it has a very different scope and focus from our work.  In contrast, DVNC showed that discretizing the results of attention or communication in GNNs can substantially improve systematic generalization.  We will update our paper to discuss connections and differences with these other works.
>
> The reviewer makes the point that discreteness in general in deep learning is not a new idea which we completely agree with. We attempted to provide a broad introduction to the issue of discrete symbols vs distributed neural networks to show how (of course) the use of discreteness is a very old idea. But, to address this point more fully, we have now improved our introduction and related works sections to include not only the papers mentioned by the Reviewer, but also discuss this literature in a systematic way.
>
> However, we would like to point out that this is not our major contribution.
>
> There are two major contributions in this work. First, our theoretical analyses show that using discrete representations for communication in machine learning models will increase a model’s robustness and decrease intrinsic dimensionality. Second, we focus on models that can be divided into modules or components and empirically show that using continuous information processing within a component but discrete communication between components improves out-of-distribution (OOD) generalization.
>
> A more detailed  point-by-point summary of our contributions are listed as follows:
>
> 1.  The theoretical analysis shows that discretization of representation decreases a model’s sensitivity and reduces the underlying dimensionality of the model.
>
> 2.  DVNC method is introduced, which conducts multi-head discretization on results of inter-component communication. The  manuscript also shows how DVNC can be incorporated into different architectures.
>
> 3.  A wide range of different experiments were conducted using DVNC in different model architectures. The results showed that discretization of results of communication among modules  improves OOD generalization in different settings.
>
> 4.  Our experimental results show that OOD generalization performance is related to the number of discretization heads (G) and codebook size (L), which agrees with our theoretical analysis.
>
> 5.  Our experimental results show that discretizing communication results is more beneficial than discretizing other parts of the model, which is why we choose to do so in this way in DVNC.
>
> We appreciate the articles suggested by the reviewer and agree that some related works are not discussed. We will update the related work section to include and discuss the articles suggested. In addition, there is a long historical literature on discreteness in neural networks going back several decades and most of them are not directly related to the current focus ,we will also  include some classic work exploring discrete neural representations including  Le et 2021(1), Lamb et al. 2019 (2) ,Amit & Brunel 1995  (3) ,Zeng, Goodman, & Smyth 1994 (4)
>
> - (1)https://arxiv.org/abs/2102.12550
> - (2)https://arxiv.org/abs/1905.11382
> - (3)https://www.tandfonline.com/doi/abs/10.1088/0954-898X_6_3_004
> - (4)https://ieeexplore.ieee.org/abstract/document/279194
>
> As for the connection between biological intelligence and machine learning, we see this paper as gaining insight from biological neural networks to try designing better artificial neural networks. We are testing one particular hypothesis: we know that the brain exploits discreteness as a representational strategy, and we are exploring whether ANNs also benefit from a qualitatively similar strategy.
>
> We thank the reviewer for pointing out the inaccurate description such as “A GNN is not composed of distinct nodes” . We will adjust the text to “DVNC discretizes communication among nodes in GNNs”. Other editing mistakes in the manuscript mentioned by the reviewer will also be corrected.

---

> > ### Comment · Reviewer_ypgC · 2021-08-18
> > **Clarification of my critisism + some more points on the theoretical analysis**
> >
> > Dear authors,
> >
> > thank you very much for your response. I appreciate a lot that you spend time to respond to my review.
> >
> > I would like to clarify that my main criticism is not the broad introduction that the paper provides or that I expect that the introduction contains results. However, I expect that a paper on discrete-valued neural communication clarifies that there are already approaches to discrete-valued neural communication and that the paper 'only' proposes a new method for discrete-valued neural communication based on VQ-VAEs, and does not invent discrete-valued neural communication itself. I pointed to three fields that already make use of discrete-valued neural communication between continuous sub-modules. Perhaps the most obvious application is in reinforcement learning. For instance, here are some examples: https://dl.acm.org/doi/10.5555/3157096.3157336, https://arxiv.org/abs/1703.04908, https://doi.org/10.1609/aaai.v34i05.6205, and https://arxiv.org/abs/2102.12550. I think these works would be interesting for the authors since these works use a very similar motivation for their work.
> >
> > However, also the other fields are relevant and there are perhaps even more. For the state-regularized RNN, the authors responded that it is different 'because it considers discretization of the hidden states of an RNN, and not the communication mechanism'. However, the proposed 'DVNC in Transformer' is also performing a discretization of the hidden states. More specifically, it performs a discretization of the output of the MultiHeadAttention(B,K,V) (see bottom of page 5) to obtain a discrete version of the output of the MultiHeadAttention, which is a discrete latent vector that is used as input for the next layer. The only difference that I see (please correct me if I am wrong) is that the current submission discretizes between layers while the state-regularized RNN discretizes between tokens/time-steps. Hence, I don't see why the DVNC in Transformer should qualify as a communication mechanism and the state-regularized RNN should not. Could you elaborate where you see a crucial difference?
> >
> > Besides this main criticism, maybe I can also add some more points on the theoretical analyses. The theoretical analyses shows that a discrete communication channel is more robust against noise since  the 'discretization process lets the communication become invariant to noise within the same category'. I am not an expert in information theory, but this seems to apply for any discrete communication channel and seems not to be specific to your work. Is this correct? And if so, robustness of  communication channels is an important topic in information theory. For instance, the noisy-channel coding theorem seems to be highly related. Do you see a connection to this theorem or other works in information theory?
> >
> > Furthermore, both Theorem 1 and 2 assume an iid draw of n examples. However, the experiments mostly focus on ood situations. Hence, I am wondering if the theorems apply in the ood experiments. If not, why are the presented theorems specifically relevant for the presented work?
> >
> > In general, Figure 4 shows that there is no systematic performance difference in the iid setting, but only in the ood experiments, which is also why the paper says that the method improves ood generalization. However, 'ood generalization' seems to be a very problematic term since the no free lunch theorem states that no method can gain an advantage in some ood settings without gaining a disadvantage in other ood settings. Or to say it in other words: wouldn't it be possible to select another set of ood experiments and to observe the exact opposite behavior (i.e. that a discrete communication channel decreases model performance)?  I would be very interested in your interpretation of this situation.

---

> > > ### Author Response · Authors · 2021-08-21
> > > **inductive bias and out-of-distribution generalization**
> > >
> > >
> > > Dear reviewer:
> > >
> > > Thank you very much for your responses.
> > >
> > > Before going  into details of each comment from the reviewer,  we like to clarify some high level logics of our work. In this study, we introduce the inductive bias of “discretization of the results of communication among different components” specifically Into  a modular model ( such as GNN, transformer and RIM).
> > >
> > > There are two parts of this inductive bias.
> > > - Part 1: we discretize representation in a model.
> > > - Part 2: While  conducting the discretization, only the results of communication among different components were discretized, representations within each component remain continuous.
> > >
> > > **However, I expect that a paper on discrete-valued neural communication clarifies that there are already approaches to discrete-valued neural communication and that the paper 'only' proposes a new method for discrete-valued neural communication based on VQ-VAEs, and does not invent discrete-valued neural communication itself. I pointed to three fields that already make use of discrete-valued neural communication between continuous sub-modules.**
> > >
> > > The authors uses the name “Discrete-valued neural communication” to describe the method that introduces both inductive bias part 1 (*discrete valued”) and part 2 (“communication”).
> > > DVNC introduced in this study is very different from other related works because 1) It introduces inductive bias part 1 and part 2. 2) The authors performed a systematic exploration of how this design decision affects OOD generalization and 3) The authors did  theoretical analysis to argue for the properties of the discretization process. The contribution comes from showing the benefit of using  DVNC as a design principle that introduces inductive bias into different modular models with quite different architectures. The points above differentiate our work from other related works and the papers referred to by the reviewers. We will include the references suggested by the reviewer into the related works section.
> > >
> > > To be more specific about inductive bias part 2, taking discretization of attention mechanisms for example, most other related work do h_new = discretize(h + attention(Q,K,V). While in DVNC, h_new = h + discretize(attention(Q,K,V)) is used, and only in communication between components.When comparing DVNC with state-regularized RNN , each model builds an internal state over time that represents a history of neural interactions. In the case of DVNC, this state (the concatenation of all module internal states) is continuous; in the case of state-regularized RNNs, this state is discrete.  What is being discretized in DVNC are the messages passed between modules, not the module states.Discrete communication is highly constrained if each of N modules is constrained to communicate in the same language. This results in N^2 constraints on the representation.  you are leveraging enforced systematicity to constrain the representation. the N for the state-regularized RNN is just 1.
> > >
> > >  **I am not an expert in information theory, but this seems to apply for any discrete communication channel and seems not to be specific to your work. Is this correct? And if so, robustness of communication channels is an important topic in information theory. For instance, the noisy-channel coding theorem seems to be highly related. Do you see a connection to this theorem or other works in information theory?.**
> > >
> > >  In this study, the authors focused exclusively on generalization in learning and that differentiates the theoretical work in this study from previous work in information theory. To the best of the authors’ knowledge,  there are no results from information theory that provide our results or similar results that show the benefits of discretization for generalization in learning. The most closely related concept with information theory would be that of information-theoretic generalization bounds [1,2]. Those do not directly show generalization in learning via the noise sensitivity term or the benefit of discretization.
> > >
> > > - [1] Information-theoretic analysis of generalization capability of learning algorithms
> > > - [2] Information-Theoretic Generalization Bounds for Stochastic Gradient Descent
> > >
> > >
> > > **“Furthermore, both Theorem 1 and 2 assume an iid draw of n examples. However, the experiments mostly focus on ood situations. Hence, I am wondering if the theorems apply in the ood experiments. If not, why are the presented theorems specifically relevant for the presented work?”**
> > >
> > > Our hypothesis is that good inductive biases help both in IID generalization in the very small-data regime and in OOD generalization. OOD generalization issues arise because the number of 'environments' seen in the in-distribution sample is too small (and would stay small even if we were to sample an infinite amount of data from the same set of environments). Hence OOD generalization is really about cross-environment generalization. Since the number of environments is typically much less than the number of examples, OOD generalization has similarities with small-sample IID generalization. To understand models behaviors in small-data regime, we conducted extra experiments on “2Dshapes” and “3D shapes” environment using only 0.1% of randomly chosen original training data for training  (<500 data points). In small data“2D shapes” experiments, baseline (GNN without DVNC) has mean MRR of 0.845 with std of 0.120 and GNN with DVNC as mean MRR of 0.983 and std of 0.028. In small data “3Dshape” experiment, baseline (GNN without DVNC) has mean MRR of 0.799 with std of 0.026 and GNN with DVNC as mean MRR of 0.952 and std of 0.050. The mean and std are obtained from 5 rounds of training/testing using different randomly picked data from the same distribution. MMR is the measurement of performance, the higher the better. These results agree with our hypothesis and will be included in the appendix section.
> > >
> > > **However, 'ood generalization' seems to be a very problematic term since the no free lunch theorem states that no method can gain an advantage in some ood settings without gaining a disadvantage in other ood settings. Or to say it in other words: wouldn't it be possible to select another set of ood experiments and to observe the exact opposite behavior (i.e. that a discrete communication channel decreases model performance)? I would be very interested in your interpretation of this situation.**
> > >
> > > Our GNN/RIMs OOD experiments are things like increasing or decreasing the number of objects, increasing the number of elements to be added and using different layouts of the game environment.  More formally what we mean by OOD, is that the system is generated by some underlying mechanisms and which can be used to induce different distributions.  For example, if we are modeling a bunch of bouncing balls, the balls move by a process like Newtonian physics.  We can change the distribution by changing the number of balls, the initial configuration of the balls, or their velocities while keeping the same underlying mechanisms in the system.  This is what we mean by OOD generalization.  We don't mean generalization to a completely arbitrary new distribution, which we agree is impossible.  We can improve the writing in the text to make this clearer.
> > >
> > > It is the DVNC inductive bias ( part 1 + part 2) we introduced that enables our model to generalize in an OOD manner. Part 1 is supported by our theoretic analysis, and part 2 is supported by our empirical experiment results in figure 5 , e,f ,g.  The model generalizes because the inductive biases we have chosen are valid across both training (set or distribution) and test (set or distribution) that we get a gain. The model will not perform well in the test ( set or distribution) where the inductive bias does not apply. Therefore, our hypothesis and results are not against the no-free-lunch theory.
> > >
> > > We thank the reviewer again for all the effort he or she put into the reviewing process that helps us make the manuscript better.

---

> ### Author Response · Authors · 2021-08-14
> **additional analysis and experiments**
>
> The authors thank the reviewer 2 again for the insightful comments and wonder if there are any additional analysis or related experiments , such as experimental comparison between DVNC and other methods mentioned, that could help verify properties of DVNC. We are open to other constructive suggestions from the reviewer as well.

---

> ### Author Response · Authors · 2021-08-24
> **Any further feedback ? Thanks :)**
>
> Dear. Reviewer,
>
> Thanks again for engaging in discussion and providing feedback.
>
> We think that the clarifications and improvements (as a result of interaction with the reviewer) has improved the presentation of our work. Since the review period is coming to an end, we would appreciate if there's any other concern of the reviewer which we can further clarify.
>
> Thanks for your time, and help. :)

---

> ### Author Response · Authors · 2021-08-28
> **further feedback appreciated**
>
> Dear reviewer, we completely understand it is a busy season but , as the clock is really counting down, we will extremely appreciate if you can let us know if you have any further concerns regarding our work and replies.  Thank you very much for your help!

---

> > ### Comment · Reviewer_ypgC · 2021-08-28
> > **No further comments**
> >
> > Dear authors,
> >
> > Thank you very much for all your responses!
> >
> > I don't have any further comments. My main point of criticism is still the fact that I think it would be inappropriate to accept a paper that makes the suggestion to invent discrete-valued neural communication even though there are already several lines of works that explore this direction. It will be up to the area chair if he/she think that this point is as severe as I see it.
> >
> > Best regards
> > the reviewer

---

> ### Author Response · Authors · 2021-08-30
> **changes of tittle and adjustment of abstract**
>
> Dear reviewer
>
> We understand the reviewer's concern that because of the title of the paper it may seem that we are inventing the discrete communication BUT we never claimed to invent discrete-valued neural communication,  Instead we show how discretizing the results of communication (especially from the attention mechanism) with a shared codebook can improve systematic and OOD generalization, which we believe is very different from the other lines of work which you have mentioned. While we may not have swayed the reviewer, we wish to acknowledge the value of the reviewer's comments on distinguishing our work from other ML models utilizing discrete representations. Regardless of the outcome of our submission to NeurIPS, we have revised our title and abstract as follow:
>
> **New title :Discretized inter-component communication in structured architectures enhances generalization**
>
>  **Adjusted abstract**:
> Deep learning has advanced from fully connected architectures to structured models organized into components, e.g., the transformer composed of positional elements, modular architectures divided into slots, and graph neural nets made up of nodes. The nature of structured models is that communication among the components is bottlenecked, typically achieved by restricted connectivity and attention. In this work, we further increase the bottlenecking effects  via discreteness of the representations transmitted between components. We hypothesize that this constraint serves as a useful form of inductive bias. Our hypothesis is motivated by past empirical work showing the benefits of discretization in non-structured architectures as well as our own theoretical results showing that discretization increases noise robustness and reduces the underlying dimensionality of the model. Building on an existing technique for discretization from the VQ-VAE, we consider multi-headed discretization with shared codebooks as the output of each architectural component.  One motivating intuition is human language in which communication occurs through multiple discrete symbols. This form of communication is hypothesized to facilitate transmission of information between functional components of the brain by providing a common interlingua, just as it does for human-to-human communication.  Our experiments show that discrete-valued inter-component communication (DVICC) substantially improves systematic generalization in a variety of architectures—transformers, modular architectures, and graph neural networks. We also show that the DVICC is robust to the choice of hyperparameters, making the method useful in practice.

---

> > ### Author Response · Authors · 2021-09-02
> > **Feedback ?**
> >
> > Dear. Reviewer,
> >
> > Does the updated title and adjusted abstract addresses the concern of the reviewer regarding the proposed work "too broad" ? Any feedback on the new title and abstract would be very useful.
> >
> > Thanks for your time, and help.

---

### Official Review · Reviewer_BL11 · 2021-07-20

**Rating:** 7
**Confidence:** 2

**Summary:**

The paper proposes a method of discretizing continuous nodes in a neural network, turning them into several discrete tokens. The method, DVNC, has theoretical and practical generalization improvements.

**Limitations And Societal Impact:**

The authors have adequately addressed the limitations and potential negative societal impact of their work.

**Main Review:**

_Originality_: The ideas proposed in this paper are novel and are centred on a generalization of the discretization node of VQ-VAE. The inspiration of VQ-VAE is clear in the paper, but oftentimes the differences are not made very clear in this work (some suggestions on _Clarity_ section).

_Quality_: The method proposed is technically sound. The authors provide a thorough theoretical analysis of the improved noise robustness of DVNC when compared to a continuous representation. The experimental procedure supports these theoretical claims. Experiments are performed in three different architectures/tasks and the results improve significantly when compared to the baseline.

_Clarity_: The paper is well-written. As briefly mentioned above, I think the authors should have used some space to better distinguish their method and VQ-VAE at the beginning of §2. A background section on that architecture would benefit the clarity of the paper.

_Significance_: The analysis and method proposed in this paper are relevant to the community and this is a line of research that warrants more future work.

__Comments__

- I think the use of $m$ as the embedding size is not consistent. It is defined as the embedding size of the discrete latent space, but then the segments $s_i$ are of size $\frac{m}{G}$ but it seems they should be of size $m$, given Eq. 1.

- In Figure 3 (right) it seems like 3 lines are missing

- Typos:

    - Line 64, 74: analyses --> analysis
    - Line 94: "selected" is repeated
    - Line 98: decent --> descent
    - Lines 194, 197, 201, 207, 210, 215, 274, 304: no space between period and next sentence, or comma has extra space/needs a space
    - Line 198: that a learned --> that are learned
    - Line 263: position --> positions

**Time Spent Reviewing:**

4h

---

> ### Author Response · Authors · 2021-08-10
> **Connections between DVNC and VQ-VAE**
>
> We appreciate the positive and insightful feedback from the reviewer. We will correct the typos in the manuscript and include an additional paragraph to clarify the differences between DVNC , VQ-VAE and other related methods. We will also update the contributions section of the manuscript to clarify how this work would benefit the community.
>
> The DVNC method proposed in this study builds on the original discretization mechanism used in VQ-VAE. The key differences are:
> 1.  We focus on models (not necessarily VAEs) that can be divided into modules or components and discretize communication among modules but keep information processing within modules continuous. We find that discretization of representations is particularly well suited to modular architectures where the components need to learn to communicate with one another and where rich information processing within a module and limited communication between modules is desired.
> 2. We construct distributed discrete representations via multi-head discretization.
> 3. Our focus on the improvement to (OOD) generalization with finite data.
>
> In short, VQ-VAE uses discretization for the purpose of learning latent variables, whereas we use a very similar discretization bottleneck but for the different purpose of imposing a bottleneck on communication between specialist modules.  Future work could explore deeper connections between DVNC and VQ-VAE.  For example, our theoretical analysis may also give insights into VQ-VAE.  A hybrid between DVNC and VQ-VAE could be used as a generative model which employs self-attention.  We hope that our work can inspire more progress in these directions.

---

> > ### Comment · Reviewer_BL11 · 2021-08-13
> > **Response to rebuttal**
> >
> > Thank you for your clarifications. I decided to keep my score.

---

### Decision · Program_Chairs · 2021-09-27

**Decision:**

Accept (Poster)

**Comment:**

This paper discusses neural network architectures that exchange discrete messages, trained using a strategy similar to the successful VQ-VAE. The paper includes both theoretical results about such architectures, as well as encouraging empirical results.

The largest concerns have to do with the discussion of the scope, as the setup discussed here has large overlap with existing but different literature. (Reviewers point out examples from reinforcement learning or on RNN hidden state discretization.) Overall, some comparison to those strategies seems possible and would improve the work, but, at the same time, the work here seems like a good step. At the very least, the connections to this others applications of discrete variables should be discussed as clearly as possible and in a positive way. (Connections here are likely to be productive.)

The paper could be strengthened with results on more realistic applications, possibly where the discrete messages can be nicely interpretable.

The authors have been very active in the discussion and have provided many clarification as well as promises for improvements of the presentation, including agreeing to a change of title that seems like it would improve clarity.